# Semi-Parametric Inducing Point Networks and Neural Processes

**Richa Rastogi,**\* **Yair Schiff, Zhaozhi Li, Ian Lee, Mert R. Sabuncu, & Volodymyr Kuleshov**\*
Cornell University
{rr568,yzs2,zl643,yl759,msabuncu,kuleshov}@cornell.edu

**Alon Hacohen**
Technion - Israel Institute of Technology
alonhacohen@campus.technion.ac.il

**Yuntian Deng**
Harvard University
dengyuntian@seas.harvard.edu

## Abstract

We introduce semi-parametric inducing point networks (SPIN), a general-purpose architecture that can query the training set at inference time in a compute-efficient manner. Semi-parametric architectures are typically more compact than parametric models, but their computational complexity is often quadratic. In contrast, SPIN attains linear complexity via a cross-attention mechanism between datapoints inspired by inducing point methods. Querying large training sets can be particularly useful in meta-learning, as it unlocks additional training signal, but often exceeds the scaling limits of existing models. We use SPIN as the basis of the Inducing Point Neural Process, a probabilistic model which supports large contexts in meta-learning and achieves high accuracy where existing models fail. In our experiments, SPIN reduces memory requirements, improves accuracy across a range of meta-learning tasks, and improves state-of-the-art performance on an important practical problem, genotype imputation.

## 1 Introduction

Recent advances in deep learning have been driven by large-scale *parametric* models (Krizhevsky et al., 2012; Peters et al., 2018; Devlin et al., 2019; Brown et al., 2020; Ramesh et al., 2022). Modern parametric models rely on large numbers of weights to capture the signal contained in the training set and to facilitate generalization (Frankle & Carbin, 2018; Kaplan et al., 2020); as a result, they require non-trivial computational resources (Hoffmann et al., 2022), have limited interpretability (Belinkov, 2022), and impose a significant carbon footprint (Bender et al., 2021).

This paper focuses on an alternative *semi-parametric* approach, in which we have access to the training set $\mathcal{D}_{\text{train}} = \{\mathbf{x}^{(i)}, \mathbf{y}^{(i)}\}_{i=1}^n$ at inference time and learn a parametric mapping $\mathbf{y} = f_\theta(\mathbf{x} \mid \mathcal{D}_{\text{train}})$ conditioned on this dataset. Semi-parametric models can *query the training set* $\mathcal{D}_{\text{train}}$ and can therefore express rich and interpretable mappings with a compact $f_\theta$. Examples of the semi-parametric framework include retrieval-augmented language models (Grave et al., 2016; Guu et al., 2020; Rae et al., 2021) and non-parametric transformers (Wiseman & Stratos, 2019; Kossen et al., 2021). However, existing approaches are often specialized to specific tasks (e.g., language modeling (Grave et al., 2016; Guu et al., 2020; Rae et al., 2021) or sequence generation (Graves et al., 2014)), and their computation scales superlinearly in the size of the training set (Kossen et al., 2021).

Here, we introduce semi-parametric inducing point networks (SPIN), a general-purpose architecture whose computational complexity at training time scales *linearly* in the size of the training set $\mathcal{D}_{\text{train}}$ and in the dimensionality of $\mathbf{x}$ and that is *constant* in $\mathcal{D}_{\text{train}}$ at inference time. Our architecture is inspired by inducing point approximations (Snelson & Ghahramani, 2005; Titsias, 2009; Wilson & Nickisch, 2015; Evans & Nair, 2018; Lee et al., 2018) and relies on a cross-attention mechanism between datapoints (Kossen et al., 2021). An important application of SPIN is in meta-learning, where conditioning on large training sets provides the model additional signal and improves accuracy,

---

\*Correspondence to Richa Rastogi and Volodymyr Kuleshov

but poses challenges for methods that scale superlinearly with $\mathcal{D}_{\text{train}}$. We use SPIN as the basis of the Inducing Point Neural Process (IPNP), a scalable probabilistic model that supports accurate meta-learning with large context sizes that cause existing methods to fail. We evaluate SPIN and IPNP on a range of supervised and meta-learning benchmarks and demonstrate the efficacy of SPIN on a real-world task in genomics—genotype imputation (Li et al., 2009). In meta-learning experiments, IPNP supports querying larger training sets, which yields high accuracy in settings where existing methods run out of memory. In the genomics setting, SPIN outperforms highly engineered state-of-the-art software packages widely used within commercial genomics pipelines (Browning et al., 2018b), indicating that our technique has the potential to impact real-world systems.

**Contributions** In summary, we introduce SPIN, a semi-parametric neural architecture inspired by inducing point methods that is the first to achieve the following characteristics:

1. Linear time and space complexity in the size and the dimension of the data during training.

2. The ability to learn a compact encoding of the training set for downstream applications. As a result, at inference time, computational complexity does not depend on training set size.

We use SPIN as the basis of the IPNP, a probabilistic model that enables performing meta-learning with context sizes that are larger than what existing methods support and that achieves high accuracy on important real-world tasks such as genotype imputation.

## 2 BACKGROUND

**Parametric and Semi-Parametric Machine Learning** Most supervised methods in deep learning are *parametric*. Formally, given a training set $\mathcal{D}_{\text{train}} = \{\mathbf{x}^{(i)}, \mathbf{y}^{(i)}\}_{i=1}^n$ with features $\mathbf{x} \in \mathcal{X}$ and labels $\mathbf{y} \in \mathcal{Y}$, we seek to learn a *fixed* number of parameters $\theta \in \Theta$ of a mapping $\mathbf{y} = f_\theta(\mathbf{x})$ using supervised learning. In contrast, non-parametric approaches learn a mapping $\mathbf{y} = f_\theta(\mathbf{x} \mid \mathcal{D}_{\text{train}})$ that can query the training set $\mathcal{D}_{\text{train}}$ at inference time; when the mapping $f_\theta$ has parameters, the approach is called *semi-parametric*. Many deep learning algorithms—including memory-augmented architectures (Graves et al., 2014; Santoro et al., 2016), retrieval-based language models (Grave et al., 2016; Guu et al., 2020; Rae et al., 2021), and non-parametric transformers (Kossen et al., 2021)—are instances of this approach, but they are often specialized to specific tasks, and their computation scales superlinearly in $n$. This paper develops scalable and domain-agnostic semi-parametric methods.

**Meta-Learning and Neural Processes** An important application of semi-parametric methods is in meta-learning, where we train a model to achieve high performance on new tasks using only a small amount of data from these tasks. Formally, consider a collection of $D$ datasets (or a meta-dataset) $\{\mathcal{D}^{(d)}\}_{d=1}^D$, each defining a task. Each $\mathcal{D}^{(d)} = (\mathcal{D}_c^{(d)}, \mathcal{D}_t^{(d)})$ contains a set of context points $\mathcal{D}_c^{(d)} = \{\mathbf{x}_c^{(di)}, \mathbf{y}_c^{(di)}\}_{i=1}^m$ and target points $\mathcal{D}_t^{(d)} = \{\mathbf{x}_t^{(di)}, \mathbf{y}_t^{(di)}\}_{i=1}^n$. Meta-learning seeks to produce a model $f(\mathbf{x}; \mathcal{D}_c)$ that outputs accurate predictions for $\mathbf{y}$ on $\mathcal{D}_t$ and on pairs $(\mathcal{D}_c, \mathcal{D}_t)$ not seen at training time. Neural Process (NP) architectures perform uncertainty aware meta-learning by mapping context sets to representations $\mathbf{r}_c(\mathcal{D}_c)$, which can be combined with target inputs to provide a distribution on target labels $\mathbf{y}_t \sim p(\mathbf{y}|\mathbf{x}_t, \mathbf{r}_c(\mathcal{D}_c))$, where $p$ is a probabilistic model. Recent successes in NPs have been driven by attention-based architectures (Kim et al., 2018; Nguyen & Grover, 2022), whose complexity scales super-linearly with context size $\mathcal{D}_c$—our method yields linear complexity. In concurrent work, Feng et al. (2023) propose a linear time method, using cross attention to reduce the size of context datasets.

**A Motivating Application: Genotype Imputation** A specific motivating example for developing efficient semi-parametric methods is the problem of genotype imputation. Consider the problem of determining the genomic sequence $\mathbf{y} \in \{\text{A}, \text{T}, \text{C}, \text{G}\}^k$ of an individual; rather than directly measuring $\mathbf{y}$, it is common to use an inexpensive microarray device to measure a small subset of genomic positions $\mathbf{x} \in \{\text{A}, \text{T}, \text{C}, \text{G}\}^p$, where $p \ll k$. Genotype imputation is the task of determining $\mathbf{y}$ from $\mathbf{x}$ via statistical methods and a dataset $\mathcal{D}_{\text{train}} = \{\mathbf{x}^{(i)}, \mathbf{y}^{(i)}\}_{i=1}^n$ of fully-sequenced individuals (Li et al., 2009). Imputation is part of most standard genome analysis workflows. It is also a natural candidate for semi-parametric approaches (Li & Stephens, 2003): a query genome $\mathbf{y}$ can normally be represented as a combination of sequences $\mathbf{y}^{(i)}$ from $\mathcal{D}_{\text{train}}$ because of the biological principle of

recombination (Kendrew, 2009), as shown in Figure 1. Additionally, the problem is a poor fit for parametric models: $k$ can be as high as $10^9$ and there is little correlation across non-proximal parts of $\mathbf{y}$. Thus, we need an unwieldy number of parametric models (one per subset of $\mathbf{y}$), whereas a single semi-parametric model can run imputation across the genome.

**Attention Mechanisms** Our approach for designing semi-parametric models relies on modern attention mechanisms (Vaswani et al., 2017), specifically *dot-product attention* $\mathrm{Att}(\mathbf{Q}, \mathbf{K}, \mathbf{V})$, which combines a query matrix $\mathbf{Q} \in \mathbb{R}^{d_q \times e_q}$ with key and value matrices $\mathbf{K} \in \mathbb{R}^{d_v \times e_q}, \mathbf{V} \in \mathbb{R}^{d_v \times e_v}$ as

$$\mathrm{Att}(\mathbf{Q}, \mathbf{K}, \mathbf{V}) = \mathrm{softmax}(\mathbf{Q}\mathbf{K}^\top / \sqrt{e_q})\mathbf{V}$$

To attend to different aspects of the keys and values, multi-head attention (MHA) extends this mechanism via $e_h$ attention heads:

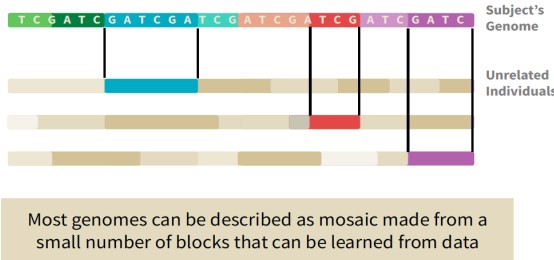

Most genomes can be described as mosaic made from a small number of blocks that can be learned from data

Figure 1: Genotype recombination

$$\mathrm{MHA}(\mathbf{Q}, \mathbf{K}, \mathbf{V}) = \mathrm{concat}(\mathbf{O}_1, ...\mathbf{O}_{e_h})\mathbf{W}^O \qquad \mathbf{O}_j = \mathrm{Att}(\mathbf{Q}\mathbf{W}_j^Q, \mathbf{K}\mathbf{W}_j^K, \mathbf{V}\mathbf{W}_j^V)$$

Each attention head projects $\mathbf{Q}, \mathbf{K}, \mathbf{V}$ into a lower-dimensional space using learnable projection matrices $\mathbf{W}_j^Q, \mathbf{W}_j^K \in \mathbb{R}^{e_q \times e_{qh}}, \mathbf{W}_j^V \in \mathbb{R}^{e_v \times e_{vh}}$ and mixes the outputs of the heads using $\mathbf{W}^O \in \mathbb{R}^{e_h e_{vh} \times e_o}$. As is commonly done, we assume that $e_{vh} = e_v/e_h$, $e_{qh} = e_q/e_h$, and $e_o = e_q$. Given two matrices $\mathbf{X}, \mathbf{H} \in \mathbb{R}^{d \times e}$, a multi-head attention block (MAB) wraps MHA together with layer normalization and a fully connected layer[1]:

$$\mathrm{MAB}(\mathbf{X}, \mathbf{H}) = \mathbf{O} + \mathrm{FF}(\mathrm{LayerNorm}(\mathbf{O})) \qquad \mathbf{O} = \mathbf{X} + \mathrm{MHA}(\mathrm{LayerNorm}(\mathbf{X}), \mathbf{H}, \mathbf{H})$$

Attention in semi-parametric models normally scales quadratically in the dataset size (Kossen et al., 2021); our work is inspired by efficient attention architectures (Lee et al., 2018; Jaegle et al., 2021b) and develops scalable semi-parametric models with linear computational complexity.

# 3 SEMI-PARAMETRIC INDUCING POINT NETWORKS

## 3.1 SEMI-PARAMETRIC LEARNING BASED ON NEURAL INDUCING POINTS

A key challenge posed by semi-parametric methods—one affecting both classical kernel methods (Hearst et al., 1998) as well as recent attention-based approaches (Kossen et al., 2021)—is the $O(n^2)$ computational cost per gradient update at training time, due to pairwise comparisons between training set points . Our work introduces methods that reduce this cost to $O(hn)$—where $h \ll n$ is a hyper-parameter—without sacrificing performance.

**Neural Inducing Points** Our approach is based on *inducing points*, a technique popular in approximate kernel methods (Wilson & Nickisch, 2015; Lee et al., 2018). A set of inducing points $\mathcal{H} = \{\mathbf{h}^{(j)}\}_{j=1}^h$ can be thought of as a "virtual" set of training instances that can replace the training set $\mathcal{D}_{\mathrm{train}}$. Intuitively, when $\mathcal{D}_{\mathrm{train}}$ is large, many datapoints are redundant—for example, groups of similar $\mathbf{x}^{(i)}$ can be replaced with a single inducing point $\mathbf{h}^{(j)}$ with little loss of information.

The key challenge in developing inducing point methods is finding a good set $\mathcal{H}$. While classical approaches rely on optimization techniques (Wilson & Nickisch, 2015), we use an *attention mechanism* to produce $\mathcal{H}$. Each inducing point $\mathbf{h}^{(j)} \in \mathcal{H}$ attends over the training set $\mathcal{D}_{\mathrm{train}}$ to select its relevant "neighbors" and updates itself based on them. We implement attention between $\mathcal{H}$ and $\mathcal{D}_{\mathrm{train}}$ in $O(hn)$ time.

**Dataset Encoding** Note that once we have a good set of inducing points $\mathcal{H}$, it becomes possible to discard $\mathcal{D}_{\mathrm{train}}$ and use $\mathcal{H}$ instead for all future predictions. The parametric part of the model makes predictions based on $\mathcal{H}$ only. This feature is an important capability of our architecture; computational complexity is now independent of $\mathcal{D}$ and we envision this feature being useful in applications where sharing $\mathcal{D}_{\mathrm{train}}$ is not possible (e.g., for computational or privacy reasons).

---

[1]We use the pre-norm parameterization for residual connections and omit details such as dropout, see Nguyen & Salazar (2019) for the full parameterization.

## 3.2 SEMI-PARAMETRIC INDUCING POINT NETWORKS

Next, we describe semi-parametric inducing point networks (SPIN), a domain-agnostic architecture with linear-time complexity.

**Notation and Data Embedding**  The SPIN model relies on a training set $\mathcal{D}_{\text{train}} = \{\mathbf{x}^{(i)}, \mathbf{y}^{(i)}\}_{i=1}^{n}$ with input features $\mathbf{x}^{(i)} \in \mathcal{X}$ and labels $\mathbf{y}^{(i)} \in \mathcal{Y}$ where $\mathcal{X}, \mathcal{Y} \in \mathcal{V}$, which is the input and output vocabulary [2]. We embed each dimension (each attribute) of $\mathbf{x}$ and $\mathbf{y}$ into an $e$-dimensional embedding and represent $\mathcal{D}_{\text{train}}$ as a tensor of embeddings $\mathbf{D} = \text{Embed}(\mathcal{D}_{\text{train}})$, $\mathbf{D} \in \mathbb{R}^{n \times d \times e}$, where $d = p + k$ is obtained from concatenating the sequence of embeddings for each $\mathbf{x}^{(i)}$ and $\mathbf{y}^{(i)}$.

The set $\mathcal{D}_{\text{train}}$ is used to learn inducing points $\mathcal{H} = \{\mathbf{h}^{(j)}\}_{j=1}^{h}$; similarly, we represent $\mathcal{H}$ via a tensor $\mathbf{H} \in \mathbb{R}^{h \times f \times e}$ of $h \leq n$ inducing points, each being a sequence of $f \leq d$ embeddings of size $e$.

To make predictions and measure loss on a set of $b$ examples $\mathcal{D}_{\text{query}} = \{\mathbf{x}^{(i)}, \mathbf{y}^{(i)}\}_{i=1}^{b}$, we use the same embedding procedure to obtain a tensor of input embeddings $\mathbf{X}_{\text{query}} \in \mathbb{R}^{b \times d \times e}$ by embedding $\{\mathbf{x}^{(i)}, \mathbf{0}\}_{i=1}^{b}$, in which the labels have been masked with zeros. We also use a tensor $\mathbf{Y}_{\text{gold}} \in \mathbb{R}^{b \times d}$ to store the ground truth labels and inputs (the objective function we use requires the model to make predictions on masked input elements as well, see below for details).

**High-Level Model Structure**  Figure 2 presents an overview of SPIN. At a high level, there are two components: (1) an Encoder module, which takes as input $\mathcal{D}_{\text{train}}$ and returns a tensor of inducing points $\mathbf{H}$; and (2) a Predictor module, which is a fully parametric model that outputs logits $\mathbf{Y}_{\text{query}}$ from $\mathbf{H}$ and $\mathbf{X}_{\text{query}}$.

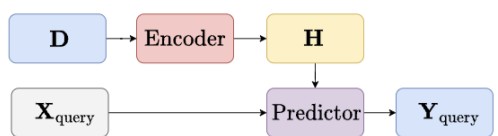

Figure 2: SPIN Model Structure

$$\mathbf{D} = \text{Embed}(\mathcal{D}_{\text{train}}) \qquad \mathbf{H} = \text{Encoder}(\mathbf{D}) \qquad \mathbf{Y}_{\text{query}} = \text{Predictor}(\mathbf{X}_{\text{query}}, \mathbf{H})$$

The encoder consists of a sequence of layers, each of which takes as input $\mathbf{D} \in \mathbb{R}^{n \times d \times e}$ and two tensors $\mathbf{H}_A \in \mathbb{R}^{n \times f \times e}$ and $\mathbf{H}_D \in \mathbb{R}^{h \times f \times e}$ and output updated versions of $\mathbf{H}_A, \mathbf{H}_D$ for the next layer. Each layer consists of a sequence of up to three cross-attention layers described below. The final output $\mathbf{H}$ of the encoder is the $\mathbf{H}_D$ produced by the last layer.

### 3.2.1 ARCHITECTURE OF THE ENCODER AND PREDICTOR

Each layer of the encoder consists of three sublayers denoted as XABA, XABD, ABLA. An encoder layer takes as input $\mathbf{H}_A, \mathbf{H}_D$ and feeds its outputs $\mathbf{H}'_A, \mathbf{H}'_D$ (defined below) into the next layer. The initial inputs $\mathbf{H}_A, \mathbf{H}_D$ of the first encoder layer are randomly initialized learnable parameters.

$$\mathbf{H}'_A = \text{XABA}(\mathbf{H}_A, \mathbf{D}) \qquad \mathbf{H}'_D = \text{XABD}(\mathbf{H}_D, \mathbf{H}'_A) \qquad \mathbf{H}_A = \text{ABLA}(\mathbf{H}'_A)$$

**Cross-Attention Between Attributes** (XABA)  An XABA layer captures the dependencies among attributes via cross-attention between the sequence of latent encodings in $\mathbf{H}$ and the sequence of datapoint features in $\mathbf{D}$.

$$\text{XABA}(\mathbf{H}_A, \mathbf{D}) = \text{MAB}(\mathbf{H}_A, \mathbf{D})$$

This updates the features of each datapoint in $\mathbf{H}_A$ to be a combination of the features of the corresponding datapoints in $\mathbf{D}$. In effect, this reduces the dimensionality of the datapoints (from $n \times d \times e$ to $n \times f \times e$). The time complexity of this layer is $O(ndfe)$, where $f$ is the dimensionality of the reduced tensor.

**Cross-Attention Between Datapoints** (XABD)  The XABD layer is the key module that takes into account the entire training set to generate inducing points.

First, it reshapes ("unfolds") its input tensors $\mathbf{H}'_A \in \mathbb{R}^{n \times f \times e}$ and $\mathbf{H}_D \in \mathbb{R}^{h \times f \times e}$ into ones of dimensions $(1 \times n \times fe)$ and $(1 \times h \times fe)$ respectively. It then performs cross-attention between

---

[2]Here we consider the case where both input and output are discrete, but our approach easily generalizes to continuous input and output spaces.

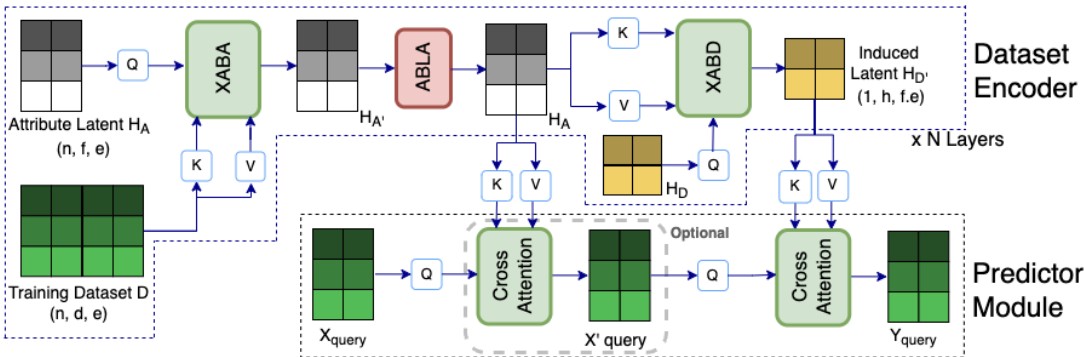

Figure 3: SPIN Architecture. Each layer of the encoder consists of sublayers XABA, ABLA and XABD, and the predictor consists of a cross attention layer. We omit feedforward layers for simplicity.

the two unfolded tensors. The output of cross-attention has dimension $(1 \times h \times fe)$; it is reshaped ("folded") into an output tensor of size $(h \times f \times e)$.

$$\text{XABD}(\mathbf{H}_D, \mathbf{H}'_A) = \text{fold}(\text{MAB}(\text{unfold}(\mathbf{H}_D), \text{unfold}(\mathbf{H}'_A))$$

This layer produces inducing points. Each inducing point in $\mathbf{H}_D$ attends to dimensionality-reduced datapoints in $\mathbf{H}'_A$ and uses its selected datapoints to update its own representation. The computational complexity of this operation is $O(nhfe)$, which is linear in training set size $n$.

**Self-Attention Between Latent Attributes** (ABLA) The third type of layer further captures dependencies among attributes by computing regular self-attention across attributes:

$$\text{ABLA}(\mathbf{H}'_A) = \text{MAB}(\mathbf{H}'_A, \mathbf{H}'_A)$$

This enables the inducing points to refine their internal representations. The dataset encoder consists of a sequence of the above layers, see Figure 3. The ABLA layers are optional based on validation performance. The input $\mathbf{H}_D$ to the first layer is part of the learned model parameters; the initial $\mathbf{H}_A$ is a linear projection of $\mathbf{D}$. The output of the encoder is the output $\mathbf{H}'_D$ of the final layer.

**Predictor Architecture** The predictor is a parametric model that maps an input tensor $\mathbf{X}_{\text{query}}$ to an output tensor of logits $\mathbf{Y}_{\text{query}}$. The predictor can use any parametric model. We propose an architecture based on a simple cross-attention operation followed by a linear projection to the vocabulary size, as shown in Figure 3:

$$\text{Predict}(\mathbf{X}_{\text{query}}, \mathbf{H}) = \text{FF}(\text{MAB}(\text{unfold}(\mathbf{X}_{\text{query}}), \text{unfold}(\mathbf{H})))$$

### 3.3 Inducing Point Neural Processes

An important application of SPIN is in meta-learning, where conditioning on larger training sets provides more information to the model, and therefore has potential to improve predictive accuracy. However, existing methods scale superlinearly with $\mathcal{D}_{\text{train}}$, and may not effectively leverage large contexts. We use SPIN as the basis of the Inducing Point Neural Process (IPNP), a scalable probabilistic model that supports fast and accurate meta-learning on large context sizes.

An IPNP defines a probabilistic model $p(\mathbf{y}|\mathbf{x}, \mathbf{r}(\mathbf{x}, \mathcal{D}_c))$ of a target variable $\mathbf{y}$ conditioned on an input $\mathbf{x}$ and a context dataset $\mathcal{D}_c$. This context is represented via a fixed-dimensional context vector $r(\mathbf{x}, \mathcal{D}_c)$, and we use the SPIN architecture to parameterize $\mathbf{r}$ as a function of $\mathcal{D}_c$. Specifically, we define $\mathbf{r}_c = \text{Encoder}(\text{Embed}(\mathcal{D}_c))$, where Encoder is the SPIN encoder, producing a tensor of inducing points. Then, we compute $\mathbf{r}(\mathbf{x}, \mathbf{r}_c) = \text{MAB}(\mathbf{x}, \mathbf{r}_c)$ via cross-attention. The model $p(\mathbf{y}|\mathbf{x}, \mathbf{r})$ is a distribution with parameters $\phi(\mathbf{x}, \mathbf{r})$, e.g., a Normal distribution with $\phi = (\mu, \Sigma)$ or a Bernoulli with $\phi \in [0, 1]$. We parameterize the mapping $\phi(\mathbf{x}, \mathbf{r})$ with a fully-connected neural network.

We further extend IPNPs to incorporate a latent variable $\mathbf{z}$ that is drawn from a Gaussian $p(\mathbf{z}|\mathcal{D}_c)$ parameterized by $\phi_{\mathbf{z}} = m(\text{Encoder}(\text{Embed}(\mathcal{D}_c)))$, where $m$ represents mean pooling across datapoints. This latent variable can be thought of as capturing global uncertainty (Garnelo et al., 2018). This results in a distribution $p(\mathbf{y}, \mathbf{z}|\mathbf{x}, \mathcal{D}_c) = p(\mathbf{y}|\mathbf{z}, \mathbf{x}, \mathcal{D}_c)p(\mathbf{z}|\mathcal{D}_c)$, where $p(\mathbf{y}|\mathbf{z}, \mathbf{x}, \mathcal{D}_c)$ is parameterized by $\phi(\mathbf{z}, \mathbf{x}, \mathbf{r}_c)$, with $\phi$ itself being a fully connected neural network. See Appendix A.6 for more detailed architectural breakdowns. Following terminology in the NP literature, we refer to our model as a conditional IPNP (CIPNP) when there is no latent variable $\mathbf{z}$ present.

## 3.4 OBJECTIVE FUNCTION

**SPIN**  We train SPIN models using a supervised learning loss $\mathcal{L}^{\text{labels}}$ (e.g., $\ell_2$ loss for regression, cross-entropy for classification). We also randomly mask attributes and add an additional loss term $\mathcal{L}^{\text{attributes}}$ that asks the model to reconstruct the missing attributes, yielding the following objective:

$$\mathcal{L}^{\text{SPIN}} = (1 - \lambda)\mathcal{L}^{\text{labels}} + \lambda\mathcal{L}^{\text{attributes}}$$

Following Kossen et al. (2021), we start with a weight $\lambda$ of 0.5 and anneal it to lean towards zero. We detail the loss terms and construction of mask matrices over labels and attrbutes in Appendix A.2. Following prior works (Devlin et al., 2019; Ghazvininejad et al., 2019; Kossen et al., 2021), we use random token level masking. Additionally, we propose chunk masking, similar to the span masking introduced in (Joshi et al., 2019), where a fraction $\rho$ of the samples selected have the mask matrix for labels $M^{(i)} = 1$, and we show the effectiveness of chunk masking in Table 5.

**IPNP**  Following the NP literature, IPNPs are trained on a meta-dataset $\{\mathcal{D}^{(d)}\}_{d=1}^D$ of context and training points $\mathcal{D}^{(d)} = (\mathcal{D}_c^{(d)}, \mathcal{D}_t^{(d)})$ to maximize the log likelihood of the target labels under the learned parametric distribution $\mathcal{L}^{\text{IPNP}} = -\frac{1}{|D|}\sum_{d=1}^D \sum_{i=1}^n \log p(\mathbf{y}_t^{(di)} \mid \mathcal{D}_c^{(d)}, \mathbf{x}_t^{(di)})$. For latent variable NPs, the objective is a variational lower bound; see Appendix A.6 for more details.

## 4 EXPERIMENTS

Semi-parametric models—including Neural Processes for meta-learning—benefit from large context sets $\mathcal{D}_c$, as they provide additional training signal. However, existing methods scale superlinearly with $\mathcal{D}_c$ and quickly run out of memory. In our experiments section, we show that SPIN and IPNP outperform state-of-the-art models by scaling to large $\mathcal{D}_c$ that existing methods do not support.

## 4.1 UCI DATASETS

We present experimental results for 10 standard UCI benchmarks, namely Yacht, Concrete, Boston-Housing, Protein (regression datasets), Kick, Income, Breast Cancer, Forrest Cover, Poker-Hand and Higgs Boson (classification datasets). We compare SPIN with Transformer baselines such as NPT (Kossen et al., 2021) and Set Transformers (Set-TF) (Lee et al., 2018). We also evaluate against Gradient Boosting (GBT) Friedman (2001), Multi Layer Perceptron (MLP) (Hinton, 1989; Glorot & Bengio, 2010), and K-Nearest Neighbours (KNN) (Altman, 1992).

Table 1: Performance Summary on UCI Datasets

|  | Traditional ML | | | Transformer | | |
|---|---|---|---|---|---|---|
|  | GBT | MLP | KNN | NPT | Set-TF | SPIN |
| Ranking ↓ | 3.00±1.76 | 4.10±1.37 | 5.44±1.01 | 2.30±1.25 | 3.63±0.92 | **2.10±0.88** |
| GPU Mem ↓ | - | - | - | 1.0x | 1.39±0.67x | **0.46±0.21x** |

Following Kossen et al. (2021), we measure the average ranking of the methods and standardize across all UCI tasks. To show the memory efficiency of our approach, we also report GPU memory usage peaks and as a fraction of GPU memory used by NPT for different splits of the test dataset in Table 1.

**Results**  SPIN achieves the best average ranking on 10 UCI datasets and uses half the GPU memory

Table 2: Effect of context size on the Poker Hand dataset

|  |  | Context Size | | | |
|---|---|---|---|---|---|
| Approach | | 4096 | 10K | 15K | 30K |
| NPT | Acc↑ | 80.11 | OOM | - | - |
|  | Mem↓ | 9.82 | OOM | - | - |
| SPIN | Acc↑ | 82.98 | 95.99 | 96.06 | 99.43 |
|  | Mem↓ | 1.73 | 3.88 | 5.68 | 10.98 |
|  | GBT | | MLP | | KNN |
| Acc↑ | 71.88 ±5.91 | | 66.09 ±9.88 | | 54.75 ±0.03 |

compared to NPT. We provide detailed results on each of the datasets and hyperparameter details in Appendix A.4. Importantly, *SPIN achieves high performance by supporting larger context sets*—we illustrate this in Table 2, where we compare SPIN and NPT on the Poker Hand dataset (70/20/10 split) using various context sizes. SPIN and NPT achieve 80-82% accuracy with small contexts, but the performance of SPIN approaches 99% as context size is increased, whereas NPT quickly runs out of GPU memory and fails to reach comparable performance.

## 4.2 NEURAL PROCESSES FOR META-LEARNING

**Experimental Setup**   Following previous work (Kim et al., 2018; Nguyen & Grover, 2022), we perform a Gaussian process meta-learning experiment, for which we create a collection of datasets $(\mathcal{D}^{(d)})_{d=1}^{D}$, where each $\mathcal{D}^{(d)}$ contains random points $(\mathbf{x}^{(di)})_{i=1}^{m}$, where $\mathbf{x}^{(di)} \in \mathbb{R}$, and target points $\mathbf{y}^{(di)} = f^{(d)}(\mathbf{x}^{(di)})$ obtained from a function $f^{(d)}$ sampled from a Gaussian Process. At each meta-training step, we sample $B = 16$ functions $\{f^{(b)}\}_{b=1}^{B}$. For each $f^{(b)}$, we sample $m \sim \mathcal{U}[\texttt{min\_ctx, max\_ctx}]$ context points and $n \sim \mathcal{U}[\texttt{min\_tgt, max\_tgt}]$ target points. The range for $n$ is fixed across all experiments at $[4, 64]$. The range for $m$ is varied from $[64, 128], [128, 256], [256, 512], [512, 1024], [1024, 2048]$. We train several different NP models for 100,000 steps and evaluate their log-likelihood on 3,000 hold out batches, with $B, m, n$ taking the same values as at training time. We evaluate conditional (CIPNP) and latent variable (IPNP) variations of our model (using $h = \frac{1}{2} \cdot \texttt{min\_ctx}$ inducing points) and compare them to other attention-based NPs: Conditional ANPs (CANP) (Kim et al., 2018), Bootstrap ANPs (BANP) (Lee et al., 2020), and latent variable ANPs (ANP) (Kim et al., 2018).

**Results**   The IPNP models attain higher performance than all baselines at most context sizes (Figure 4). Interestingly, IPNPs generalize better—recall that IPNPs are more compact models with fewer parameters, hence are less likely to overfit. We also found that *increased context size led to improved performance* for all models; however, baseline NPs required excessive resources, and BANPs ran out of memory entirely. In contrast, IPNPs scaled to large context sizes using up to 50% less resources.

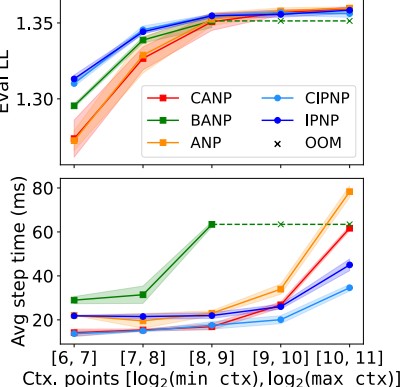

## 4.3 GENOTYPE IMPUTATION

Genotype imputation is the task of inferring the sequence $\mathbf{y}$ of an entire genome via statistical methods from a small subset of positions $\mathbf{x}$—usually obtained from an inexpensive DNA microarray device (Li et al., 2009)—and a dataset $\mathcal{D}_{\text{train}} = \{\mathbf{x}^{(i)}, \mathbf{y}^{(i)}\}_{i=1}^{n}$ of fully-

Figure 4: Inducing Point NPs outperform NP baselines and train much faster. Plots display mean $\pm$ std. deviation from 5 runs with different random seeds.

sequenced individuals (Li et al., 2009). Imputation is part of most standard workflows in genomics (Lou et al., 2021) and involves mature imputation software (Browning et al., 2018a; Rubinacci et al., 2020) that benefits from over a decade of engineering (Li & Stephens, 2003). These systems are fully non-parametric and match genomes in $\mathcal{D}_{\text{train}}$ to $\mathbf{x}, \mathbf{y}$; their scalability to modern datasets of up to millions of individuals is a known problem in the field (Maarala et al., 2020). Improved imputation holds the potential to reduce sequencing costs and improve workflows in medicine and agriculture.

**Experiment Setup**   We compare against one of the state-of-the-art packages, Beagle (Browning et al., 2018a), on the 1000 Genomes dataset (Clarke et al., 2016), following the methodology described in Rubinacci et al. (2020). We use 5008 complete sequences $\mathbf{y}$ that we divide into train/val/test splits of 0.86/0.12/0.02, respectively, following Browning et al. (2018b). We construct inputs $\mathbf{x}$ by masking positions that do not appear on the Illumina Omni2.5 array (Wrayner). Our experiments in Table 3 focus on five sections of the genome for chromosome 20. We pre-process the input into sequences of $K$-mers for all methods (see Appendix A.3). The performance of this task is measured via the Pearson correlation coefficient $R^2$ between the imputed SNPs and their true value at each position.

We compare against NPTs, Set Transformers, and classical machine learning methods. NPT-16, SPIN-16 and Set Transformer-16 refer to models using an embedding dimension 16, a model depth of 4, and one attention head. NPT-64, SPIN-64 and Set Transformer-64 refer to models using an embedding dimension of 64, a model depth of 4, and 4 attention heads. SPIN uses 10 inducing points for datapoints ($h$=10, $f$=10). A batch size of 256 is used for Transformer methods, and we train using the lookahead Lamb optimizer (Zhang et al., 2019).

Table 3: Performance Summary on Genomic Sequence Imputation. ($*$) represents parametric models. A difference of 0.5% is statistically significant at pvalue 0.05.

|  | GBT* | MLP* | KNN | Beagle | NPT-16 | Set-TF-16 | SPIN-16 |
|---|---|---|---|---|---|---|---|
| Pearson $R^2$ ↑ | 87.63 | 95.31 | 89.70 | 95.64 | 95.84 ±0.06 | **95.97±0.09** | 95.92 ±0.12 |
| Param Count↓ | - | 65M | - | - | 16.7M | 33.4M | **8.1M** |

**Results** Table 3 presents the main results for genotype imputation. Compared to the previous state-of-the-art commercial software, Beagle, which is specialized to this task, all Transformer-based methods achieve strong performance, despite making fewer assumptions and being more general. While all the three Transformer-based approaches report similar Pearson $R^2$, SPIN achieves competitive performance with a much smaller parameter count. Among traditional ML approaches, MLP perform the best, but requires training one model per imputed SNP, and hence cannot scale to full genomes. We provide additional details on resource usage and hyper-parameter tuning in Appendix A.3.

## 4.4 SCALING GENOTYPE IMPUTATION VIA META-LEARNING

One of the key challenges in genotype imputation is making predictions for large numbers of SNPs. To scale to larger sets of SNPs, we apply a meta-learning based approach, in which a single shared model is used to impute arbitrary genomic regions.

**Experimental Setup** We create a meta-training set $\{\mathcal{D}^{(d)}\}_{d=1}^{D}$, where each $(\mathcal{D}_c^{(d)}, \mathcal{D}_t^{(d)})$ corresponds to one of $D$ independent genomic segments, $\mathcal{D}_c^{(d)}$ is the set of reference genomes in that segment, and $\mathcal{D}_t^{(d)}$ is the set of genomes that we want to learn to impute. At each meta-training step, we sample a new pair $(\mathcal{D}_c^{(d)}, \mathcal{D}_t^{(d)})$ and update the model parameters to maximize the likelihood of $\mathcal{D}_t^{(d)}$. We further create three independent versions of this experiment—denoted Full, 50%, and 25%—in which the segments defining $(\mathcal{D}_c^{(d)}, \mathcal{D}_t^{(d)})$ contain 400, 200, and 100 SNPs respectively. We fit an NPT and a CIPNP model parameterized by SPIN-64 architecture and apply chunk-level masking method instead of token-level masking.

**Results** Table 4 shows that both the CIPNP (SPIN-64) and the NPT-64 model support the meta-learning approach to genotype imputation and achieve high performance, with CIPNP being more accurate. We provide performance for each region within the datasets in Appendix A.3, Table 8. However, the NPT model cannot handle full-length genomic segments and runs out of memory on the full experiment. This again highlights the ability of SPIN to scale and thus solve problems that existing models cannot.

Table 4: Multiple Windows Experiment

| | | 25% | 50% | Full |
|---|---|---|---|---|
| NPT-64 | $R^2$ ↑ | 95.06 | 92.89 | OOM |
| | Mem ↓ | 12.36 | 19.86 | OOM |
| SPIN-64 | $R^2$ ↑ | 95.38 | 93.55 | 93.90 |
| | Mem ↓ | 5.33 | 8.30 | 16.44 |

**Masking** Table 5 shows the effect of chunk style masking over token level masking for SPIN in order to learn the imputation algorithm. As the genomes are created by copying over chunks due to the biological principle of recombination, we find that chunk style masking of labels at train time provides significant improvements over random token level masking for the meta learning genotype imputation task.

Table 5: Masking

| Masking | $R^2$ ↑ |
|---|---|
| Token | 80.48 |
| Chunk | 95.32 |

**Ablation Analysis** To evaluate the effectiveness of each module, we perform ablation analysis by gradually removing components from SPIN. We remove components one at a time and compare the performance with default SPIN configuration. In Table 6, we observe that for the genomics dataset (SNPs 424600-424700) and UCI Boston Housing (BH) dataset, both XABD and XABA are crucial components. We discuss ablation with a synthetic experiment setup in Appendix A.7

Table 6: Ablation Analysis

|         | GEN ↑ | BH ↓       |
|---------|-------|------------|
| SPIN    | 94.05 | 3.0 ±0.6   |
| -XABD   | 93.50 | 3.1 ±0.8   |
| -XABA   | 93.89 | 3.2 ±1.5   |

## 5 RELATED WORK

**Non-Parametric and Semi-Parametric Methods** Non-parametric methods include approaches based on kernels (Davis et al., 2011), such as Gaussian processes (Rasmussen, 2003) and support vector machines (Hearst et al., 1998). These methods feature quadratic complexity (Bach, 2013), which motivates a long line of approximate methods based on random projections (Achlioptas et al., 2001), Fourier analysis (Rahimi & Recht, 2007), and inducing point methods (Wilson et al., 2015). Inducing points have been widely applied in kernel machines (Nguyen et al., 2020), Gaussian processes classification (Izmailov & Kropotov, 2016), regression (Cao et al., 2013), semi-supervised learning (Delalleau et al., 2005), and more (Hensman et al., 2015; Tolstikhin et al., 2021).

**Deep Semi-Parametric Models** Deep Gaussian Processes (Damianou & Lawrence, 2013), Deep Kernel Learning (Wilson et al., 2016), and Neural Processes (Garnelo et al., 2018) build upon classical methods. Deep GPs rely on sophisticated variational inference methods (Wang et al., 2016), making them challenging to implement. Retrieval augmented transformers (Bonetta et al., 2021) use attention to query external datasets in specific domains such as language modeling (Grave et al., 2016), question answering (Yang et al., 2018), and reinforcement learning (Goyal et al., 2022) and in a way that is similar to earlier memory-augmented models (Graves et al., 2014). Non-Parametric Transformers (Kossen et al., 2021) use a domain-agnostic architecture based on attention that runs in $O(n^2 d^2)$ at training time and $O(nd^2)$ at inference time, while ours runs in $O(nd)$ and $O(d)$, respectively.

**Attention Mechanisms** The quadratic cost of self-attention (Vaswani et al., 2017) can be reduced using efficient architectures such as sparse attention (Beltagy et al., 2020), Set Transformers (Lee et al., 2018), the Performer (Choromanski et al., 2020), the Nystromer (Xiong et al., 2021), Long Ranger (Grigsby et al., 2021), Big Bird (Zaheer et al., 2020), Shared Workspace (Goyal et al., 2021), the Perceiver (Jaegle et al., 2021b;a), and others (Katharopoulos et al., 2020; Wang et al., 2020). Our work most closely resembles the Set Transformer (Lee et al., 2018) and Perceiver (Jaegle et al., 2021b;a) mechanisms—we extend these mechanisms to cross-attention between datapoints and use them to attend to datapoints, similar to Non-Parametric Transformers (Kossen et al., 2021).

**Set Transformers** Lee et al. (2018) introduce inducing point attention (ISA) blocks, which replace self-attention with a more efficient cross-attention mechanism that maps a set of $d$ tokens to a new set of $d$ tokens. In contrast, SPIN cross-attention compresses sets of size $d$ into smaller sets of size $h < d$. Each ISA block also uses a different set of inducing points, whereas SPIN layers iteratively update the same set of inducing points, resulting in a smaller memory footprint. Finally, while Set Transformers perform cross-attention over features, SPIN performs cross-attention between datapoints.

## 6 CONCLUSION

In this paper, we introduce a domain-agnostic general-purpose architecture, the semi-parametric inducing point network (SPIN) and use it as the basis for Induced Point Neural Process (IPNPs). Unlike previous semi-parametric approaches whose computational cost grows quadratically with the size of the dataset, our approach scales linearly in the size and dimensionality of the data by leveraging a cross attention mechanism between datapoints and induced latents. This allows our method to scale to large datasets and enables meta learning with large contexts. We present empirical results on 10 UCI datasets, a Gaussian process meta learning task, and a real-world important task in genomics, genotype imputation, and show that our method can achieve competitive, if not better, performance relative to state-of-the-art methods at a fraction of the computational cost.

## 7 ACKNOWLEDGMENTS

This work was supported by Tata Consulting Services, the Cornell Initiative for Digital Agriculture, the Hal & Inge Marcus PhD Fellowship, and an NSF CAREER grant (#2145577). We would like to thank Edgar Marroquin for help with preprocessing of raw genomic data. We would like to thank NPT authors - Jannic and Neil for helpful discussions and correspondence regarding NPT architecture. We would also like to thank the anonymous reviewers for their significant effort to provide suggestions and helpful feedback, thereby improving our paper.

## 8 REPRODUCIBILITY

We provide details on the compute resources in Appendix A.1, including GPU specifications. Code and data used to reproduce experimental results are provided in Appendix C. We provide error bars on the reported results by varying seeds or a different test split, however for certain large datasets, such as UCI datasets for Kick, Forest Cover, Protein, Higgs and Genomic Imputation experiments with large output sizes, we reported results on a single run due to computational limitations. These details are provided in Appendix A.3, Appendix A.4 and Appendix A.5.

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

APPENDIX: SEMI-PARAMETRIC INDUCING POINT NETWORKS AND NEURAL PROCESSES

# A EXPERIMENTAL DETAILS

## A.1 COMPUTE RESOURCES

We use 24GB NVIDIA GeForce RTX 3090, Tesla V100-SXM2-16GB and NVIDIA RTX A6000-48GB GPUs for experiments in this paper. A result is reported as OOM if it did not fit in the 24GB GPU memory. We do not use multi-GPU training or other memory-saving techniques such as gradient checkpointing, pruning, mixed precision training, etc. but note that these are orthogonal to our approach and can be used to further reduce the computational complexity.

## A.2 TRAINING OBJECTIVE

We define a binary mask matrix for a given sample $i$ as $M^{(i)} = [m_1^{(i)}, m_2^{(i)}, \cdots m_l^{(i)}]$, where $l = k$ for labels and $l = p$ for attributes. Then the loss over labels and attributes for each sample $i$ is given by,

$$\mathcal{L}^{\text{labels},(i)}(\mathbf{y}_{\text{pred}}^{(i)}, \mathbf{y}_{\text{true}}^{(i)}, M^{\text{labels},(i)}) = \sum_{j=1}^{k} m_j^{(i)} \mathcal{L}(\mathbf{y}_{\text{pred},j}^{(i)}, \mathbf{y}_{\text{true},j}^{(i)})$$

$$\mathcal{L}^{\text{attributes},(i)}(\mathbf{x}_{\text{pred}}^{(i)}, \mathbf{x}_{\text{true}}^{(i)}, M^{\text{attributes},(i)}) = \sum_{j=1}^{p} m_j^{(i)} \mathcal{L}(\mathbf{x}_{\text{pred},j}^{(i)}, \mathbf{x}_{\text{true},j}^{(i)})$$

where $\mathcal{L}(\mathbf{y}_{\text{pred},j}^{(i)}, \mathbf{y}_{\text{true},j}^{(i)}) = -\sum_{c=1}^{\text{C}} \mathbf{y}_{\text{true},j,c}^{(i)} \log(\text{softmax}(\mathbf{y}_{\text{pred},j,c}^{(i)}))$ Cross Entropy Loss for C-way Classification and $\mathcal{L}(\mathbf{y}_{\text{pred},j}^{(i)}, \mathbf{y}_{\text{true},j}^{(i)}) = (\mathbf{y}_{\text{true},j}^{(i)} - \mathbf{y}_{\text{pred},j}^{(i)})^2$ for MSE Loss. $\mathcal{L}(\mathbf{x}_{\text{pred},j}^{(i)}, \mathbf{x}_{\text{true},j}^{(i)})$ for attributes that are reconstructed is computed analogously

For chunk masking, a fraction $\rho$ of the samples selected have the mask matrix for labels $M^{(i)} = 1$

$$M^{(i)} = \begin{cases} 1, & \text{with probability } \rho \\ 0, & \text{otherwise} \end{cases}$$

## A.3 GENOMIC SEQUENCE IMPUTATION

Imputation is performed on single-nucleotide polymorphisms (SNPs) with a corresponding marker panel specifying the microarray. We randomly sample five sections of the genome for chromosome 20 for conducting experiments. Each section is selected with 100 SNPs to be predicted and 150 closest SNPs are obtained. For compact encoding of SNPs, we form $K$-mers, which are commonly used in various genomics applications (Compeau et al., 2011), where $K$ is a hyper-parameter that controls the granularity of tokenization (how many nucleotides are treated as a single token). This now becomes a $2^K$-way classification task. We set $K$ to 5 for all the genomics experiments, so that there are 20 (100/5) target SNPs to be imputed and 30 (150/5) attributes per sampled section. We report pearson $R^2$ for each of the five sections in Table 7 with error bars per window for five different seeds. For computational load, we report peak GPU memory usage in GB where applicable, an average of train time per epoch in seconds, and parameter count per method. Table 8 provides Pearson $R^2$ for each of the 10 regions using a single model, thus learning the Genotype imputation algorithm.

In Table 9, we analyze the effect of increasing reference haplotypes during training on pearson $R^2$ computed by NPT and SPIN. The reference haplotypes in the train dataset are gradually increased from a small fraction of 1% to 100% available. Pearson $R^2$ is reported cumulatively for 10 randomly selected regions with window size=300. We observe that the performance for both SPIN and NPT improves with increasing reference dataset. However, NPT cannot be used beyond a certain set of reference samples due to its GPU memory footprint, while SPIN yields improved performance.

**Hyperparameters** In Table 10, we provide the range of hyper-parameters that were grid searched for different methods. Beagle is a specialized software using dynamic programming and does not require any hyper-parameters from the user.

Table 7: Performance on Genomics Imputation

| Approach | | Pearson $R^2 \uparrow$ | Peak GPU Mem (GB) $\downarrow$ | Params Count $\downarrow$ | Avg. Train time/epoch(s) $\downarrow$ |
|---|---|---|---|---|---|
| Genomics Dataset (SNPs 68300-68400) | | | | | |
| Traditional ML | GBT | 81.12 | - | - | - |
| | MLP | 97.63 | - | - | - |
| | KNN | 86.96 | - | - | - |
| Bio Software | Beagle | 98.07 | - | - | - |
| Transformer | NPT-16 | 96.96 ±0.28 | 0.45 | 16.7M | 2.22 |
| | STF-16 | 97.02 ±0.23 | 0.76 | 33.4M | 2.93 |
| | SPIN-16 | 97.13 ±0.21 | 0.28 | 8.1M | 2.23 |
| Genomics Dataset (SNPs 169500-169600) | | | | | |
| Traditional ML | GBT | 91.53 | - | - | - |
| | MLP | 97.19 | - | - | - |
| | KNN | 95.65 | - | - | - |
| Bio Software | Beagle | 97.87 | - | - | - |
| Transformer | NPT-16 | 97.44 ±0.08 | 0.45 | 16.7M | 1.98 |
| | STF-16 | 98.07 ±0.15 | 0.76 | 33.4M | 2.99 |
| | SPIN-16 | 97.50 ±0.12 | 0.28 | 8.1M | 2.76 |
| Genomics Dataset (SNPs 287600-287700) | | | | | |
| Traditional ML | GBT | 81.12 | - | - | - |
| | MLP | 96.20 | - | - | - |
| | KNN | 95.56 | - | - | - |
| Bio Software | Beagle | 92.62 | - | - | - |
| Transformer | NPT-16 | 97.07 ±0.06 | 0.45 | 16.7M | 2.24 |
| | STF-16 | 97.09 ±0.07 | 0.76 | 33.4M | 2.99 |
| | SPIN-16 | 97.11 ±0.04 | 0.28 | 8.1M | 2.60 |
| Genomics Dataset (SNPs 424600-424700) | | | | | |
| Traditional ML | GBT | 82.77 | - | - | - |
| | MLP | 91.98 | - | - | - |
| | KNN | 84.39 | - | - | - |
| Bio Software | Beagle | 93.72 | - | - | - |
| Transformer | NPT-16 | 93.49 ±0.70 | 0.45 | 16.7M | 2.23 |
| | STF-16 | 93.38 ±0.90 | 0.76 | 33.4M | 2.90 |
| | SPIN-16 | 93.83 ±0.65 | 0.28 | 8.1M | 2.21 |
| Genomics Dataset (SNPs 543000-543100 ) | | | | | |
| Traditional ML | GBT | 72.66 | - | - | - |
| | MLP | 89.56 | - | - | - |
| | KNN | 78.22 | - | - | - |
| Bio Software | Beagle | 94.58 | - | - | - |
| Transformer | NPT-16 | 91.30 ±1.14 | 0.45 | 16.7M | 2.35 |
| | STF-16 | 91.49 ±0.39 | 0.76 | 33.4M | 2.52 |
| | SPIN-16 | 91.36 ±0.18 | 0.28 | 8.1M | 2.28 |

Table 8: Pearson $R^2 \uparrow$ for each region for the Multiple Windows Experiment

| Runs | Small | | Medium | | Large | |
|---|---|---|---|---|---|---|
| | SPIN | NPT | SPIN | NPT | SPIN | NPT |
| 1 | 97.34 | 96.85 | 94.30 | 93.86 | 97.55 | OOM |
| 2 | 98.09 | 97.80 | 83.82 | 81.70 | 90.38 | - |
| 3 | 96.98 | 97.13 | 97.04 | 96.91 | 87.46 | - |
| 4 | 93.39 | 93.61 | 95.46 | 95.24 | 95.93 | - |
| 5 | 92.12 | 91.66 | 93.19 | 92.53 | 97.05 | - |
| 6 | 95.59 | 94.39 | 94.39 | 93.99 | 91.30 | - |
| 7 | 94.58 | 93.86 | 89.97 | 88.63 | 93.66 | - |
| 8 | 93.34 | 92.67 | 93.70 | 93.39 | 94.48 | - |
| 9 | 85.01 | 85.82 | 94.30 | 93.54 | 94.39 | - |
| 10 | 97.70 | 97.31 | 95.52 | 94.80 | 87.94 | - |
| Total | 95.39 | 95.06 | 93.55 | 92.89 | 93.90 | OOM |

Table 9: Cumulative Pearson $R^2 \uparrow$ for 10 randomly selected genomic windows

| Reference Samples | Pearson $R^2 \uparrow$ | | Peak GPU (GB) | |
|---|---|---|---|---|
| | SPIN | NPT | SPIN | NPT |
| 44 (1%) | 84.87 | 85.00 | 8.64 | 18.07 |
| 219 (5%) | 86.25 | 85.54 | 8.77 | 18.43 |
| 658 (15%) | 87.55 | 86.35 | 9.1 | 19.69 |
| 1316 (30%) | 90.33 | - | 9.59 | OOM |
| 4388 (100%) | 92.91 | - | 12.83 | OOM |

Table 10: Hyperparameters for Genomics Dataset

| Model | Hyperparameter | Setting |
|---|---|---|
| NPT, SPIN, Set Transformer | Embedding Dimension | $[16, 128]$ |
| | Depth | $[2, 8]$ |
| | Label Masking | $[0, 0.5]$ |
| | Target Masking | $[0.3]$ |
| | Learning rate | $[1e-5, 1e-2]$ |
| | Dropout | $[0.4, 0.6]$ |
| | Batch Size | $[256, 5008$ (No Batching)$]$ |
| | Inducing points | $[3, 100]$ |
| Gradient Boosting | Max Depth | $[5, 10]$ |
| | n_estimators | $[100]$ |
| | Learning rate | $[1e-2]$ |
| MLP | Hidden Layer Sizes | $[(500, 500, 500)]$ |
| | Batch Size | $[128, 256]$ |
| | L2 regularization | $[0, 1e-2]$ |
| | Learning rate init | $[1e-4, 1e-2]$ |
| KNN | n_neighbors | $[2, 1000]$ |
| | weights | $[distance]$ |
| | algorithm | $[auto]$ |
| | Leaf Size | $[10, 100]$ |
| Bio Software | None | None |

## A.4 UCI REGRESSION TASKS

In Table 11, we report results for 10 cross-validation (CV) splits for Yacht and Concrete datasets, 5 CV splits for Boston-Housing datasets, and 1 CV split for Protein dataset. Number of splits were chosen according to computational requirements. Below we provide details about each dataset.

- **Yacht** dataset consists of 308 instances, 1 continuous, and 5 categorical features.
- **Boston Housing** dataset consists of 506 instances, 11 continuous, and 2 categorical features.
- **Concrete** consists of 1030 instances, and 9 continuous features.
- **Protein** consists of 45,730 instances, and 9 continuous features.

Table 11: Performance on UCI Regression Datasets

| Approach | | RMSE ↓ | Peak GPU Mem (GB) ↓ | Params Count ↓ | Avg. Train time/epoch(s) ↓ |
|---|---|---|---|---|---|
| Boston-Housing | | | | | |
| Traditional ML | GBT | 3.44±0.22 | - | - | - |
| | MLP | 3.32±0.39 | - | - | - |
| | KNN | 4.27±0.37 | - | - | - |
| Transformer | NPT | 2.92±0.15 | 8.2 | 168.0M | 1.45 |
| | STF | 3.33±1.73 | 16.5 | 336.0M | 1.99 |
| | SPIN | 3.01 ±0.55 | 6.3 | 127.2M | 1.63 |
| Yacht | | | | | |
| Traditional ML | GBT | 0.87±0.37 | - | - | - |
| | MLP | 0.83±0.18 | - | - | - |
| | KNN | 11.97±2.06 | - | - | - |
| Transformer | NPT | 1.42±0.64[3] | 2.1 | 42.7M | 0.10 |
| | STF | 1.29±0.34 | 4.1 | 85.4M | 0.19 |
| | SPIN | 1.28±0.66 | 1.6 | 32.2M | 0.07 |
| Concrete | | | | | |
| Traditional ML | GBT | 4.61±0.72 | - | - | - |
| | MLP | 5.29±0.74 | - | - | - |
| | KNN | 8.62±0.77 | - | - | - |
| Transformer | NPT | 5.21±0.20 | 3.4 | 69.9M | 0.13 |
| | STF | 5.35±0.80 | 6.8 | 139.9M | 0.21 |
| | SPIN | 5.17±0.87 | 1.9 | 39.4M | 0.21 |
| Protein | | | | | |
| Traditional ML | GBT | 3.61 | - | - | - |
| | MLP | 3.62 | - | - | - |
| | KNN | 3.77 | - | - | - |
| Transformer | NPT | 3.34 | 13.1 | 86.1M | 18.13 |
| | STF | 3.39 | 5.3 | 172.3M | 8.34 |
| | SPIN | 3.31 | 3.2 | 43.0M | 24.28 |

## A.5 UCI CLASSIFICATION TASKS

In Table 12, we report results for 10 CV splits for Breast Cancer dataset and 1 CV split for Kick, Income, Forest Cover, Poker-Hand and Higgs Boson datasets. Number of splits were chosen according to computational requirements. Below we provide details about each dataset.

---

[3]NPT reports a mean of 1.27 on this task that we could not reproduce. However, we emphasize that for UCI experiments, all the model parameters are kept same for all the transformer methods.

- **Breast Cancer** dataset consists of 569 instances, 31 continuous features, and 2 target classes.
- **Kick** dataset consists of 72,983 instances, 14 continuous and 18 categorical features, and 2 target classes.
- **Income** consists of 299,285 instances, 6 continuous and 36 categorical features, and 2 target classes.
- **Forest Cover** consists of 581,012 instances, 10 continuous and 44 categorical features, and 7 target classes.
- **Poker-Hand** consists of 1,025,010 instances, 10 categorical features, and 10 target classes.
- **Higgs Boson** consists of 11,000,000 instances, 28 continuous features, and 2 target classes.

We provide the range of hyperparameters for UCI datasets in Table 13. Additionally, we provide average ranking separated by Regression and Classification tasks in Table 14 and Table 15, respectively.

Table 12: Performance on UCI Classification Datasets

| Approach | | Accuracy ↑ | Peak GPU Mem (GB) ↓ | Params Count ↓ | Avg. Train time/epoch(s) ↓ |
|---|---|---|---|---|---|
| Breast Cancer | | | | | |
| Traditional ML | GBT | 94.03±2.74 | - | - | - |
| | MLP | 94.03±3.05 | - | - | - |
| | KNN | 95.26±2.48 | - | - | - |
| Transformer | NPT | 95.79±1.22 | 2.6 | 51.3M | 0.15 |
| | STF | 94.91±1.53 | 5.2 | 102.6M | 0.21 |
| | SPIN | 96.32±1.54 | 0.9 | 16.9M | 0.18 |
| Kick | | | | | |
| Traditional ML | GBT | 90.20 | - | - | - |
| | MLP | 89.96 | - | - | - |
| | KNN | 87.71 | - | - | - |
| Transformer | NPT | 90.04 | 14.9 | 232.6M | 56.22 |
| | STF | 90.03 | 15.0 | 465.0M | 52.35 |
| | SPIN | 90.06 | 3.6 | 73.7M | 27.76 |
| Income | | | | | |
| Traditional ML | GBT | 95.8 | - | - | - |
| | MLP | 95.4 | - | - | - |
| | KNN | 94.8 | - | - | - |
| Transformer | NPT | 95.6 | 24 | 1504M | - |
| | STF | - | OOM | - | - |
| | SPIN | 95.6 | 11.5 | 418.9M | 68.02 |
| Forest Cover | | | | | |
| Traditional ML | GBT | 96.70 | - | - | - |
| | MLP | 95.20 | - | - | - |
| | KNN | 90.70 | - | - | - |
| Transformer | NPT | 96.73 | 18.0 | 644.7M | 230.47 |
| | STF | - | OOM | - | - |
| | SPIN | 96.11 | 5.4 | 162.7M | 138.38 |
| Poker-Hand | | | | | |
| Traditional ML | GBT | 78.71 | - | - | - |
| | MLP | 56.40 | - | - | - |
| | KNN | 54.75 | - | - | - |
| Transformer | NPT | 80.11 | 9.8 | 104.0M | 93.56 |
| | STF | 79.89 | 3.1 | 52.1M | 83.13 |
| | SPIN | 82.98 | 1.7 | 11.8M | 72.05 |
| Higgs Boson | | | | | |
| Traditional ML | GBT | 76.50 | - | - | - |
| | MLP | 78.30 | - | - | - |
| | KNN | - | - | - | - |
| Transformer | NPT | 80.70 | 14.7 | 179.5M | 1,569.39 |
| | STF | 80.48 | 12.8 | 359.0M | 1,796.94 |
| | SPIN | 80.01 | 4.9 | 62.1M | 983.44 |

Table 13: Hyperparameters for UCI Dataset

| Model | Hyperparameter | Setting |
|---|---|---|
| NPT, SPIN, Set Transformer | Embedding Dimension | $[16, 128]$ |
| | Depth | $[8]$ |
| | Label Masking | $[0, 0.5]$ |
| | Target Masking | $[0.3]$ |
| | Learning rate | $[1e-5, 1e-2]$ |
| | Dropout | $[0.4, 0.6]$ |
| | Batch Size | [2048, No Batching] |
| | Inducing points | [5,10] |
| Gradient Boosting | Max Depth | $[3, 10]$ |
| | n_estimators | $[50, 1000]$ |
| | Learning rate | $[1e-3, 0.3]$ |
| MLP (Boston Housing Breast Cancer, Concrete, and Yacht) | Hidden Layer Sizes | $[(25)-(500), (25, 25)-(500, 500),$ $(25, 25, 25)-(500, 500, 500)]$ |
| | Batch Size | $[32, 256]$ |
| | L2 regularization | $[0, 1]$ |
| | Learning rate | [constant, invscaling, adaptive] |
| | Learning rate init | $[1e-5, 1e-1]$ |
| MLP (Kick, Income) | Hidden Layer Sizes | $[(25, 25, 25)-(500, 500, 500)]$ |
| | Batch Size | $[128, 256]$ |
| | L2 regularization | $[0, 1e-2]$ |
| | Learning rate | [constant, invscaling, adaptive] |
| | Learning rate init | $[1e-5, 1e-1]$ |
| KNN (Boston Housing Breast Cancer, Concrete, and Yacht) | n_neighbors | $[2, 100]$ |
| | weights | [uniform, distance] |
| | algorithm | [ball_tree, kd_tree, brute] |
| | Leaf Size | $[10, 100]$ |
| KNN (Kick, Income) | n_neighbors | $[2, 1000]$ |
| | weights | [distance] |
| | algorithm | [auto] |
| | Leaf Size | $[10, 100]$ |

Table 14: Average Ranking on UCI Regression Dataset (Yacht, Boston Housing, Concrete, Protein) based on RMSE

| Approach | | Average Ranking order ↓ | Peak GPU Mem (relative to NPT)↓ |
|---|---|---|---|
| Traditional ML | GBT | 3.00±1.83 | - |
| | MLP | 3.25±1.71 | - |
| | KNN | 6.00±0.00 | - |
| Transformer | NPT | 2.75±1.71 | 1.0x |
| | STF | 4.00±0.82 | 1.71±0.48x |
| | SPIN | 2.00±0.82 | 0.65±0.09x |

Table 15: Average Ranking on UCI Classification Dataset (Breast Cancer, Kick, Income, Forest Cover, Poker-Hand, Higgs-Boson based on Classification Accuracy

|  | Approach | Average Ranking order ↓ | Peak GPU Mem (relative to NPT)↓ |
|---|---|---|---|
| Traditional ML | GBT | 3.00±1.90 | - |
|  | MLP | 4.67±0.82 | - |
|  | KNN | 5.00±1.22 | - |
| Transformer | NPT | 2.00±0.89 | 1.0x |
|  | STF | 3.25±0.96 | 1.05±0.70x |
|  | SPIN | 2.17±0.98 | 0.31±0.10x |

## A.6 NEURAL PROCESSES

**Variational Lower Bound** The variational lower bound objective used to train latent NPs is as follows:

$$\mathcal{L}^{\text{IPNP,ELBO}} = -\frac{1}{|D|} \sum_{d=1}^{D} \Big[ \sum_{i=1}^{n} \log p_\theta(\mathbf{y}_t^{(di)} \mid \mathbf{z}, \mathcal{D}_c^{(d)}, \mathbf{x}_t^{(di)}) + \text{KL}(q(\mathbf{z} \mid \mathcal{D}_t^{(d)}, \mathcal{D}_c^{(d)}) \parallel p(\mathbf{z} \mid \mathcal{D}_c^{(d)})) \Big]$$

where KL is the Kullback–Leibler divergence, $q(\mathbf{z} \mid \mathcal{D}_t^{(d)}, \mathcal{D}_c^{(d)})$ is the posterior distribution conditioned on target and context sets, and $p(\mathbf{z} \mid \mathcal{D}_c^{(d)})$ is the prior conditioned only on the context.

**Hyperparameters** Learning rates for all experiments were set to $5e^{-4}$ with a Cosine Annealing learning rate scheduler applied. Model parameters were optimized using the ADAM optimizer (Kingma & Ba, 2014).

**Architectures** In Table 16 and 17, we detail the architecture for the conditional NPs (CANP, BANP, CIPNP) and latent variable NPs (ANP, IPNP) used in Section 4.2, respectively. Note that although the conditional NPs do not have a latent path, in order to make them comparable in terms of number of parameters we add another deterministic encoding *Pooling Encoder* to these models, as described in Lee et al. (2020). In these tables, we remark where XABD is used as opposed to regular self attention between context data points. We use the following shorthand notation below: $\mathbf{X}_c$ are context features for a batch of datasets stacked into a tensor of size $B \times m \times 1 \times 1$, and $\mathbf{X}_t$ is defined similarly. $\mathbf{D}$ denotes the full dataset, features and labels for both context and target, stacked into a tensor of size $B \times m + n \times 2 \times 1$.

Note, although the equations described in Section 3.3 and in Table 16 and 17 use tensors of order 4, in practice we use tensors of order 3 and permute the dimensions of the tensor in order to ensure that attention is performed along the correct dimension (i.e., data points).

Finally, in Figure 5, we provide a more detailed diagram of the (conditional) IPNP architecture, which excludes the additional *Pooling Encoder*.

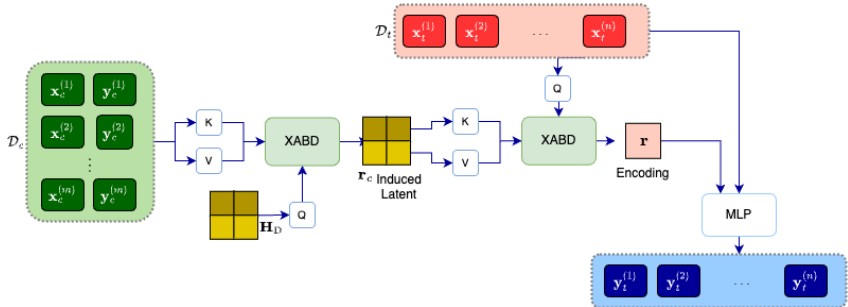

Figure 5: CIPNP architecture diagram

Table 16: CANP / BANP / CIPNP architecture (no latent path)

| Input | Layer(s) | Output |
|---|---|---|
| *Cross Attention Encoder* | | |
| $\mathbf{X}_t \in \mathbb{R}^{B \times n \times 1 \times 1}$ | MLP (1 hidden layer, ReLU activation) | $\mathbf{Q} \in \mathbb{R}^{B \times n \times 128 \times 1}$ |
| $\mathbf{X}_c \in \mathbb{R}^{B \times m \times 1 \times 1}$ | MLP (1 hidden layer, ReLU activation) | $\mathbf{K} \in \mathbb{R}^{B \times m \times 128}$ |
| $\mathbf{D} \in \mathbb{R}^{B \times m \times 2 \times 1}$ | (1) MLP (1 hidden layer, ReLU activation), | |
| | (2) $\mathrm{MAB}(\mathbf{D}, \mathbf{D})$ (Self attn between data points) | |
| | for CANP/BANP | |
| | $\mathrm{MAB}(\mathbf{H}_D, \mathbf{D})$ (XABD) for CIPNP | $\mathbf{V} = \mathbf{r}_c \in \mathbb{R}^{B \times m \times 128 \times 1}$ |
| $\mathbf{Q}, \mathbf{K}, \mathbf{V}$ | Cross attn between query and context points | $\mathbf{r} \in \mathbb{R}^{B \times n \times 128 \times 1}$ |
| *Pooling Encoder* | | |
| $\mathbf{D} \in \mathbb{R}^{B \times m \times 2 \times 1}$ | (1) MLP (1 hidden layer, ReLU activation), | |
| | (2) $\mathrm{MAB}(\mathbf{D}, \mathbf{D})$ (Self attn between data points) | |
| | for CANP/BANP | |
| | $\mathrm{MAB}(\mathbf{H}_D, \mathbf{D})$ (XABD) for CIPNP | |
| | (3) Mean pooling (on context points) | |
| | (4) MLP (1 hidden layer, ReLU activation) | |
| | (5) Repeat $n$ times | $\mathbf{r}' \in \mathbb{R}^{B \times n \times 128 \times 1}$ |
| *Decoder* | | |
| $\mathtt{concat}(\mathbf{X}_t, \mathbf{r}, \mathbf{r}')$ | (1) FC | |
| $\in \mathbb{R}^{B \times n \times 257 \times 1}$ | (2) MLP (2 hidden layer, ReLU activation) | $\phi \in \mathbb{R}^{B \times n \times 2}$ |
| $\phi$ | $\mathtt{chunk}$ (splits input into 2 tensors of equal size) | $\mu, \Sigma \in \mathbb{R}^{B \times n \times 1}$ |
| $\mu, \Sigma$ | Sampler | $\mathbf{Y}_t \in \mathbb{R}^{B \times n \times 1}$ |
| | | $\sim \mathcal{N}(\mu, \Sigma^2)$ |

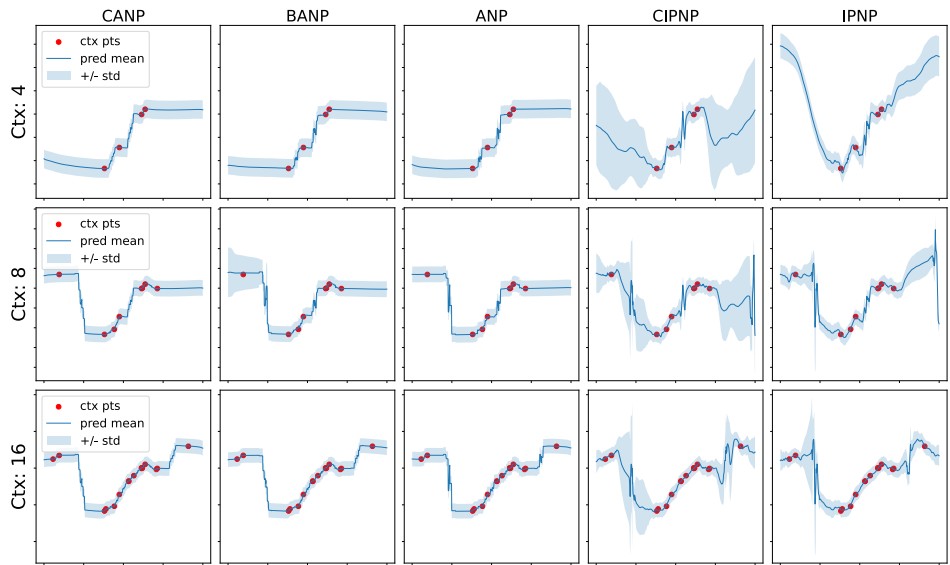

Figure 6: Predicted mean ± standard deviation of **y** for different NP models given varying context sizes: 4 (*top*), 8 (*middle*), and 16 (*bottom*).

Table 17: ANP / IPNP architecture (latent path)

| Input | Layer(s) | Output |
|---|---|---|
| *Cross Attention Encoder* | | |
| | Same as CANP / BANP / CIPNP (Table 16) | |
| *Latent Pooling Encoder* | | |
| $\mathbf{D} \in \mathbb{R}^{B \times m \times 2 \times 1}$ | (1) MLP (1 hidden layer, ReLU activation), | |
| | (2) $\mathrm{MAB}(\mathbf{D}, \mathbf{D})$ (Self attn between data points) | |
| | for ANP | |
| | $\mathrm{MAB}(\mathbf{H}_D, \mathbf{D})$ (XABD) for IPNP | |
| | (3) Mean pooling (on context points) | |
| | (4) MLP (1 hidden layer, ReLU activation) | $\phi'_{\mathbf{z}} \in \mathbb{R}^{B \times 256}$ |
| $\phi_{\mathbf{z}}$ | chunk (splits input into 2 tensors of equal size) | $\mu_{\mathbf{z}}, \Sigma_{\mathbf{z}} \in \mathbb{R}^{B \times 128 \times 1}$ |
| $\mu_{\mathbf{z}}, \Sigma_{\mathbf{z}}$ | (1) Sampler | |
| | (2) Repeat $n$ times | $\mathbf{z} \in \mathbb{R}^{B \times n \times 128 \times 1}$ |
| | | $\sim \mathcal{N}(\mu_{\mathbf{z}}, \Sigma_{\mathbf{z}}^2)$ |
| *Decoder* | | |
| $\mathrm{concat}(\mathbf{X}_t, \mathbf{r}, \mathbf{z})$ | (1) FC | |
| $\in \mathbb{R}^{B \times n \times 257 \times 1}$ | (2) MLP (2 hidden layer, ReLU activation) | $\phi \in \mathbb{R}^{B \times n \times 2}$ |
| $\phi$ | chunk (splits input into 2 tensors of equal size) | $\mu, \Sigma \in \mathbb{R}^{B \times n \times 1}$ |
| $\mu, \Sigma$ | Sampler | $\mathbf{Y}_t \in \mathbb{R}^{B \times n \times 1}$ |
| | | $\sim \mathcal{N}(\mu, \Sigma^2)$ |

**Qualitative Uncertainty Estimation Results** In Figure 6, we show baseline and inducing point NP models trained with context sizes $\in [64, 128]$ and display the output of these models on new datasets with varying numbers of context points (4, 8, 16). We observe that the CIPNP and IPNP models better capture uncertainty in regions where context points have not been observed.

**Quantitative Calibration Results** To provide more quantitative results of how well our NP models capture uncertainty relative to baselines, we take models trained with context sizes $\in [64, 128]$ and evaluate them on 1,000 evaluation batches each with number of targets points ranging from 4 to 64. We repeat this experiment three times with varying numbers of context points (4, 8, 16) available for

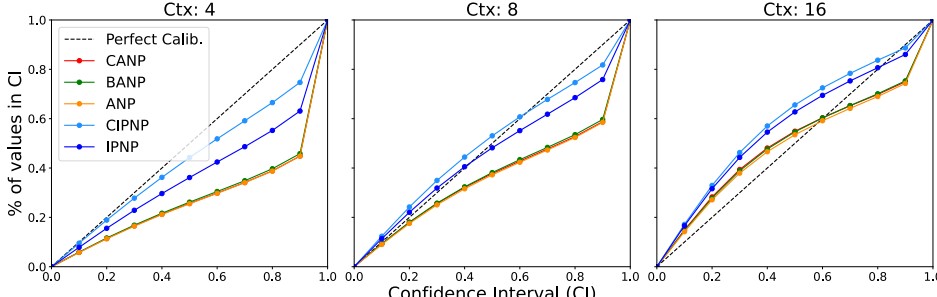

Figure 7: Calibration of NP models given varying context sizes: 4 (*left*), 8 (*middle*), and 16 (*right*).

Table 18: Calibration scores ($\downarrow$) for NP models across context sizes. Calibration score equals mean squared deviation from $45°$ line of number of target points falling within confidence intervals ranging from 0 to 1 by intervals of 0.1, see Equation (1). Best (lowest) scores for each context size are bolded.

| | Calibration score $\downarrow$ | | | | |
|---|---|---|---|---|---|
| Context | CANP | BANP | ANP | CIPNP | IPNP |
| 4 | 0.065 | 0.062 | 0.065 | **0.006** | 0.023 |
| 8 | 0.025 | 0.024 | 0.026 | **0.002** | 0.004 |
| 16 | 0.006 | **0.005** | 0.005 | 0.011 | 0.008 |

each evaluation batch. In Figure 7, we see that in lower context regimes, CIPNP and IPNP models are better calibrated than the other baselines. As context size increases, the calibration of all the models deteriorates. This is further reflected in Table 18, where we display model calibration scores. Letting $CI$ be confidence intervals ranging from 0 to 1.0 by intervals of 0.1, $p_{CI}$ be the fraction of target labels that fall within confidence interval $CI$, and $n$ be the number of confidence intervals, this calibration score is equal to:

$$\frac{1}{n} \sum_{CI=0}^{1} (p_{CI} - CI)^2 \tag{1}$$

This score measures deviation of each model's calibration plot from the $45°$ line. Future work will explore the mechanisms that enable inducing point models to better capture uncertainty.

### A.7 ABLATION ANALYSIS WITH SYNTHETIC EXPERIMENT

We formulate a synthetic experiment where the model can only learn via XABD layers. First, we initialize a random binary matrix with 50% probability of 1's, number of rows=5000, and number of columns=50. We set the last 20 columns to be target labels. Next, we copy a section of the dataset and divide it into three equal and disjoint parts for train query, val query, and test query. Since there is no correlation between the features and the target, the only way for the model to learn is via XABD layers (for a small dataset the model can also memorize the entire training dataset). This is similar to the synthetic experiment in NPT (Kossen et al., 2021), except that there is no relation between the features and target in our setup. We find that both default SPIN and SPIN with XABD only component achieves 100% binary classification accuracy, whereas SPIN with XABA only component achieves 70.01% classification accuracy, indicating the effectiveness of XABD component.

### A.8 QUALITATIVE ANALYSIS FOR CROSS-ATTENTION

In order to understand what type of inducing points are learnt by the latent $H_D$, we formulate a toy synthetic dataset as shown in Figure 8. We start with two centroids consisting of binary strings with 120 bits and add bernoulli noise with $p = 0.1$. We create the labels as another set of 4 bit binary strings and apply bernoulli noise with $p = 0.1$. In this way we create a dataset with datapoints belonging to two separate clusters. Figure 8 (a) shows the projection of this dataset with two principal

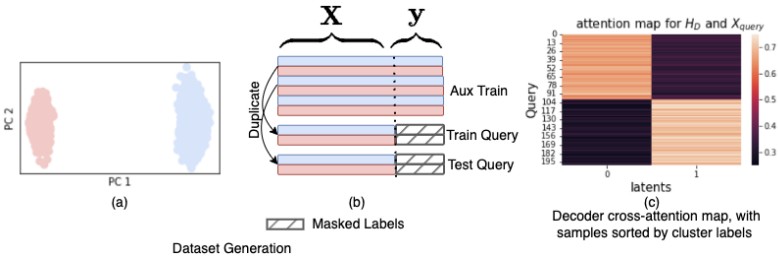

Figure 8: Synthetic Experiment analyzing cross-attention: (a)-(b) Data generating process to form two distinct clusters and data matrix with duplicate query samples and their labels masked. (c) Cross-attention map between query samples and the latent $H_D$

components when projected with PCA, highlighting that the dataset consists of two distinct clusters. We duplicate a part of this data to form the query samples so that they can be looked up from the latent via the cross attention mechanism. Figure 8 (b) shows the schematic of dataset with masked values for query labels. We use a SPIN model with a single XABD module, two induced latents ($h$=2) and input embedding dimension of 32 and inspect the cross-attention mechanism of the decoder. In Figure 8 (c), we plot the decoder cross attention map between test query and the induced latent and observe that the grouped query datapoints attend to the two latents consistent with the data generating clusters.

## A.9 IMAGE CLASSIFICATION EXPERIMENTS

We conducted additional experiments comparing SPIN and NPT on two image classification datasets. Following NPT, we compare the results for image classification task using a linear patch encoder for MNIST and CIFAR10 dataset (which is the reason for the lower accuracies compared to using CNN-based encoders). Table 19 shows that for the linear patch encoder, SPIN and NPT both perform similarly in terms of accuracy, but SPIN uses far fewer parameters.

Table 19: Image Classification Experiments

| Dataset | Approach | Classification Accuracy ↑ | Peak GPU (GB) | Params Count |
|---------|----------|---------------------------|---------------|--------------|
| MNIST | NPT | 97.92 | 1.33 | 33.34M |
| | SPIN | 97.70 | 0.37 | 8.68M |
| CIFAR-10 | NPT | 68.20 | 18.81 | 900.36M |
| | SPIN | 68.81 | 5.13 | 217.53M |

## A.10 EFFECT OF NUMBER OF INDUCED POINTS

We conducted sensitivity analysis of SPIN's performance with respect to $h$ and $f$ and found that SPIN is fairly robust to the choice of these hyper-parameters, as evidenced by the low standard deviations in Table 20. This reflects redundancy in data and why attending to the entire dataset is inefficient.

Table 20: Effect of induced points $h, f$ for one genomic window (SNPs 424600-424700)

| Induced Points $h$ | Induced Points $f$ | Pearson $R^2$ ↑ | Peak GPU (GB) |
|--------------------|--------------------|-----------------|---------------|
| $\{5 \cdots 30\}$ | 10 | 94.03 $\pm$0.50 | 0.28 $\pm$0. |
| 10 | $\{5 \cdots 30\}$ | 94.15 $\pm$0.25 | 0.32 $\pm$0.05 |
| $\{5 \cdots 30\}$ | $\{5 \cdots 30\}$ | 94.03 $\pm$0.28 | 0.32 $\pm$0.05 |

## B COMPLEXITY ANALYSIS

We provide time complexity for one gradient step, with $n_l$ as the number of layers, batch size $b$ equal to training dataset size $n$ during training, and one query sample during inference for transformer methods in Table 21. There are two operations that contribute heavily to the time complexity. First is

computation of $Q.K^T$, second is the four times expansion in the feedforward layers. For NPT, the time complexity is given by maximum of ABD, ABA, and four times expansion in feedforward layers during ABD, that is $\max(n_l n^2 de, n_l nd^2 e, 4n_l nd^2 e^2)$ during training and inference. Set Transformer consists of ISAB blocks that perform one cross-attention between latent and dataset to project into smaller space and a cross attention between dataset and latent to project back into input space for each layer. This results in complexity that is $\max(2n_l ndfe, 2n_l nhfe, 8n_l nd^2 e^2)$ during training and inference. For SPIN, the time complexity is given by maximum of XABD, XABA, ABLA, four times expansion in feedforward layers and one cross-attention for Predictor module. This can be formulated as $\max(n_l nhfe, n_l ndfe, n_l nf^2 e, 4n_l nf^2 e, nhde, 4nd^2 e^2)$. At inference, SPIN only uses the Predictor module, with the resultant complexity as $\max(hde, 4d^2 e^2)$.

We note that during training for NPT, if $n > 4de$, then $Q.K^T$ computation in ABD dominates, otherwise the four times expansion of feedforward for ABD dominates. For Set Transformer, usually the four times expansion of feedforward dominates. For SPIN, depending on values for $n_l, d, f, h, e$, different computations can dominate, however it is always linear in dataset size $n$. During inference SPIN's time complexity is independent of number of layers $n_l$ and dataset size $n$ and depends entirely on inducing datapoints $h$, model embedding dimension $e$, and feature+target space $d$.

Table 21: Time Complexity

| Approach | Time Complexity | |
| --- | --- | --- |
| | Train | Test |
| NPT | $\max(n_l n^2 de, n_l nd^2 e, 4n_l nd^2 e^2)$ | $\max(n_l n^2 de, n_l nd^2 e, 4n_l nd^2 e^2)$ |
| STF | $\max(2n_l ndfe, 2n_l nhfe, 8n_l nd^2 e^2)$ | $\max(2n_l ndfe, 2n_l nhfe, 8n_l nd^2 e^2)$ |
| SPIN | $\max(n_l nhfe, n_l ndfe, n_l nf^2 e, 4n_l nf^2 e, nhde, 4nd^2 e^2)$ | $\max(hde, 4d^2 e^2)$[4] |

## C  CODE AND DATA AVAILABILITY

**UCI and Genomic Task Code**   The experimental results for UCI and genomic task can be reproduced from here.

**Neural Processes Code**   The experimental results for the Neural Processes task can be reproduced from here.

**Data for Genomics Experiment**   The vcf file containing genotypes can be downloaded from 1000Genomes chromosome 20 vcf file. Additionally, the microarray used for genomics experiment can be downloaded from HumanOmni2.5 microarray. Beagle software, used as baseline, can be obtained from Beagle 5.1.

**UCI Datasets**   All UCI datasets can be obtained from UCI Data Repository.

---

[4]The complexity at test time for SPIN is $\max(nhde, 4nd^2 e^2)$ when either using the optional cross-attention in the predictor module or when the encoder is enabled at test time, such as in the multiple windows genomic experiment.

