# OpenReview forum: "Semi-Parametric Inducing Point Networks and Neural Processes"
_ICLR.cc/2023/Conference — ICLR 2023 poster_

### Official Review · Reviewer_wVgc · 2022-10-25

**Confidence:** 3
**Correctness:** 4
**Technical Novelty And Significance:** 4
**Empirical Novelty And Significance:** 4
**Recommendation:** 8

**Clarity, Quality, Novelty And Reproducibility:**

The paper is well written and easy to follow.

The experiments are thorough and high quality.

Using inducing points in the context of NPT is novel to my knowledge, and the authors have done a good job building upon this idea by including large scale meta-learning experiments.

Reproducibility of the method is ensured by a thorough appendix and provided code.

Question for the authors: How do you explain the performance gains in Table 2 over NPT when the context size is identical? Intuitively, it seems like the bottleneck introduced by inducing points should adversely impact performance.

**Strength And Weaknesses:**

Strengths:
 - The paper is well motivated, as it addresses a clear weakness of NPT in a suitable and convincing manner.
 - The method is clearly explained and sensibly constructed.
 - The experimental evaluation is thorough and highlights important aspects of the method, including its sensitivity to context size, its parameter count, and its memory usage.
 - The improvements over NPT on UCI and meta learning experiments are quite convincing.

Weaknesses:
 - On the genome imputation experiment, there is no significant performance increase over prior methods, the result is only notable in terms of SPIN's generality and low parameter count. Maybe some more concretely desirable property such as inference time could be measured here for a more convincing argument.
 - The resolution of the figures, especially figure 3, should be improved, ideally by making them a vector graphics.

**Summary Of The Paper:**

The paper proposes a new semi-parametric learning method called SPIN. It builds upon non-parametric transformers (NPT), which use attention between training points to learn compact and effective predictors at the cost of quadratic runtime in the number of training points considered (context size). SPIN addresses this by utilizing inducing points, a small constant number of vectors summarizing the large number of training points. The resulting architecture scales linearly in the context size at training time, and linearly only in the selected number of inducing points at test time. It is demonstrated that depending on the selected context size, SPIN yields improved prediction performance, reduced computational cost, or both. Additionally, SPIN is employed for metalearning on synthetic and genome imputation data, where it delivers improved/competitive performance, respectively.

**Summary Of The Review:**

In summary, I think the paper represents a clear step forward for semi-parametric methods. The idea of using inducing points is well executed and thoroughly evaluated. The meta-learning and genome imputation experiments push the applicability of such methods to new domains. I therefore recommend acceptance.

-----

Post Rebuttal Update: After reading the other reviews and the authors' responses, I maintain my positive view of the paper.

---

> ### Author Response · Authors · 2022-11-19
> **Response to Reviewer wVGC**
>
> We thank the reviewer for their detailed feedback. We address the reviewer's concerns below.
>
> **Concern 1: The performance improvements of SPIN on genotype imputation are not well-explained.**
>
> While SPIN performs similarly to baselines such as NPT with small context set sizes (i.e., small reference set sizes), it significantly outperforms existing methods at larger context set sizes. Moreover, existing methods are not able to match the performance of SPIN because they run out of memory when using similarly large contexts.
>
> We provide the following experiment where we analyze the effect of increasing reference haplotypes during training on pearson $R^2$ computed by NPT and SPIN. The reference haplotypes in the train dataset are gradually increased from a small fraction of 1\% to 100\% available. Pearson $R^2$ is reported cumulatively for 10 randomly selected regions with window size=300. As the size of genomic biobanks increase, there is a growing need for methods that can scale.
> We observe that the performance for both SPIN and NPT improves with increasing the reference dataset. However, NPT cannot be used beyond a certain set of reference samples due to its GPU memory footprint.
>
>
> |  Reference Samples|| 1% (44)|5% (219)|15% (658)|30% (1316)|100% (4388)|
> |:-:|:-:|:-:|:-:|:-:|:-:|:-:|
> |  Pearson $R^2$ $\uparrow$| SPIN|84.87|86.25|87.55|90.33|92.91|
> |  | NPT |85.00|85.54|86.35|-|-|
> | GPU Mem(GB)| SPIN|8.64|8.77|9.10|9.59|12.83|
> |  | NPT|18.07|18.43|19.69|OOM|OOM|
>
> **Additional minor comments**
>
> - In general, our claim is that for the same context size, SPIN performs competitively with NPT. Empirically, we do see slight gains at the same context size for certain datasets,  as noted in Table 2, however these gains at the same context size are often not statistically significant and not consistent across datasets. As mentioned by the reviewer, we show that the performance of SPIN improves over NPT as the context increases. This improvement is shown empirically for multiple datasets (UCI experiment, meta learning and genomic sequence imputation experiment) and is clearly statistically significant.
> - We have improved the figure resolution and added them as vectorized figures.

---

### Official Review · Reviewer_TnrL · 2022-11-03

**Confidence:** 4
**Correctness:** 4
**Technical Novelty And Significance:** 2
**Empirical Novelty And Significance:** 3
**Recommendation:** 6

**Clarity, Quality, Novelty And Reproducibility:**

The authors write relatively clearly here and the work is empirically of high quality.

The writing is also of good quality, and the authors put great attention to detail into listing lots of the many details in their architecture, but by its own nature this work feels somewhat engineered and is not easy to deeply understand from a mathematical principles point of view, but rather as a list of specific implementations that has a desired effect. This is partly the nature of the work, but I would hope for more understanding why these techniques perform well.
What the authors do a great job on is explaining the memory and computational footprint of this architecture, which is also a key aspect of its ability to scale. I came away with a good understanding of why this would scale well on larger context sets thanks to the writing.

I was appreciative of the authors' great job to position this work fairly among newer and older literature, the scholarship here is thorough and fair and spans all walks of ML where inducing points and meta-learning/neural processes have been seen.

With respect to reproducibility, I do believe I could reimplement the key layers here after reading the paper and the authors share lots of detail in their appendix. By its nature -again- work like this probably depends strongly on the details of the experimental tuning, I came away with the impression the authors are handling this well.

**Strength And Weaknesses:**

Strengths:
The technique the authors propose appears to work well empirically.
While the core setup is a familiar one of neural processes, the particular handling of inducing points here allows the models to remain competitive and scale beyond the typical comparators, which is a promising avenue for its future use as one may want to incorporate larger context datasets D_c.

The authors do a good job comparing to a few benchmarks such as UCI datasets, GP-prior modeling, and genotype imputation.

Weaknesses:
Some brief ablation experiments seem to indicate that the lion;s share of the work is carried by the cross-data attention modeling part here. It would be great to get deeper insight into this.
In general the work appears to propose an architecture which feels somewhat "engineered" and while the authors show that it works, it would be great to understand better why each part works, how sequencing these different layers makes sense , what calculation the model is really amortizing over here, and deepen the analysis of the ablations.
What does the ABLA layer really do? I understand the authors claim it refines H_A, but it is not clear to me what this is amortizing and the work does not really convey the intuition here.


**Summary Of The Paper:**

The authors introduce a mechanism for semi-parametric prediction using neural attention mechanisms and a variant of inducing points.

In this work, the authors combine ideas such as cross-attention between attributes and between datapoints, yielding a framework -denoted SPIN- which given a new dataset produces a set of lower dimensional inducing points H in an encoder structure.
This set H, can either be used to update to the next set H given data, or it can be used in a prediction layer in a comnstruct which allows performing predictions of the style p(y|x, H).

The authors propose to then utilize SPIN for a neural process variant, which using these inducing points is able to incorporate large context sizes in context datasets D_c to make predictions on query datasets.

In their experiments, they show competitive performance on various tasks related to the neural process literature where they exhibit strong scalability due to better memory utilization of their method compared to useful baselines.
Of particular note is an example for genotype imputation, which seems to benefit strongly in this meta-learning scenario from the proposed methodology.

**Summary Of The Review:**

The authors present an evolution and marriage of the streams of work on neural processes and cross-datapoint attention, which they bring together to propose an architecture which can utilize large context sets to perform semi-parametric prediction.
I really enjoyed the key application to genotype imputation as an example that would be uniquely enabled by this model.

Overall the paper is of good enough quality and interest and although it does not meaningfully extend or propose much technical novelty or deep insights into why the utilized pieces here work as well as they do, the empirical qualities regarding scalability and positioning of the paper appear and useful and valuable.

If the authors were able to better explain and understand the interplay of their layers and which computations they are amortizing I would have been more excited.

---

> ### Author Response · Authors · 2022-11-19
> **Response to Reviewer TnRL**
>
> We thank the reviewer for their detailed feedback. We address the reviewer's concerns below.
>
> **Concern 1: The need for additional insight into the architecture, specifically the XABD and ABLA layers.**
>
> *Response 1.1. Intuition behind the XABD layer.*
>
> Intuitively, SPIN tries to "compress" a large dataset $\mathcal{D}$ into a smaller set of inducing points $\mathcal{H}$. Often, many datapoints in $\mathcal{D}$ will be similar to each other, and thus redundant. Inducing point methods replace such redundant datapoints with a single inducing point. Classical inducing point methods rely on an optimization procedure to find $\mathcal{H}$. In contrast, SPIN uses the XABD layer to compute $\mathcal{H}$ from $\mathcal{D}$ using cross-attention. The XABD layer takes as input a tensor of inducing points $\mathbf{H}$ and a tensor of datapoints $\mathbf{D}$; it outputs a new tensor of inducing points $\mathbf{H}'$ that are formed when $\mathbf{H}$ attends to $\mathbf{D}$; each inducing point in $\mathbf{H}$ queries $\mathbf{D}$, attends to a group of similar datapoints, and updates its representation in a way that summarizes the signal contained in this group of datapoints.
>
> In order to better explain the workings of the XABD module, we perform the following experiment, which demonstrates that the above intuition is correct (see Appendix A.8).
>   - We start with two centroids consisting of binary strings with 120 bits ($X$) and add bernoulli noise with $p=0.1$.
>   - We create the labels ($y$) to be predicted as a different set of 4 bit binary strings and apply bernoulli noise ($p=0.1$). In this way, we create a dataset with datapoints belonging to two separate clusters.
>   - Figure 8 (a) shows the projection of the 120-bit string features with the first two principal components from PCA, highlighting that the dataset consists of two clusters.
>   - We duplicate a part of this data to form the query points so that they can be "looked up" from the latent via the cross attention mechanism.
>   - Figure 8 (b) shows the schematic of dataset with masked values for query labels.
>   - We use a SPIN model with a single XABD module, two induced latents ($h=2$) and input embedding dimension of 32 and inspect post-hoc, the cross-attention mechanism of the decoder.
>   - In Figure 8 \(c\), we plot the decoder cross attention map between test query and the induced latent. We observe that the grouped query datapoints attend to the two latents consistent with the data generating clusters.
>
>
> *Response 1.2. Intuition behind the ABLA layer.*
>
> We draw inspiration for the ABLA module from existing literature, where self-attention is performed on projected latents. ABLA similarly performs self-attention on the $H_A$ latent and thus has low computational cost. Our reasoning for including the ABLA was to provide a general purpose architecture that could be used to benefit various datasets.

---

### Official Review · Reviewer_GJez · 2022-11-03

**Confidence:** 3
**Correctness:** 3
**Technical Novelty And Significance:** 3
**Empirical Novelty And Significance:** 3
**Recommendation:** 6

**Clarity, Quality, Novelty And Reproducibility:**

The paper is very well written overall, aside from some explanation of the design choices and intuitions.

It's hard for me to speak on novelty since I'm not very familiar with the background work. Overall, it gave me impressions of the perceiver model (mentioned in related work), but applied to encode the entire training set.

The experiments seem nice, but again I don't know the standard procedure for semi-parametric model evaluations.

Code and data availability section in the Appendix is nice.

**Strength And Weaknesses:**

Strengths:

This paper is excellently written. I enjoyed reading it - particularly the intro and background, and it flows nicely.

Overall, the results are slightly better than existing methods. But SPIN requires significantly less GPU usage and param count than existing methods, as well as allows for larger context size. Computational cost grows linearly in the training set size compared to  the quadratic growth of existing methods.

The experiments seem convincing, and I particularly like the genomic imputation experiments. This type of model seems like a good method for genomics, where we always have the reference genome and prior lab-experiments.


Weaknesses:

The motivation for why we need more GPU efficient methods compared to NPT isn't very clear. Table 2 shows the effect of a larger context size that NPT can't fit in memory, but aside from that I didn't see anything else. For something like genomic imputations, those aren't time-sensitive so it shouldn't be big deal waiting a little longer or using a little more compute.

The paper has a lot of equations, but it boils down to a fairly simple method using multi-head cross-attention. I appreciated the detail, but I think the paper could be greatly simplified.

The related work section lists a lot of papers, but doesn't provide much of a comparison to the proposed method.

What is the intuition of "folding" in XABD? I think this was a key part of the paper, but it wasn't motivated. Similarly, what is the intuition for ABLA?

Minor:

What are "attributes" in XABA? I found this nomenclature confusing.

Figures should be vectorized.

I would move the "Attention Mechanisms" subsection to an appendix (aside from explaining what MAB is).

Tables and Figures could be organized a little bit better. Hard to read in the text when it's flowing in every direction around the figures and tables.


**Summary Of The Paper:**

This paper introduces semi-parametric inducing point networks, which learn a latent representation of the entire training set which can be queried at inference time efficiently. The authors experiment on the UCI benchmarks, a synthetic meta-learning task, and a genomic imputation task. The key contribution is a decrease in computational cost vs existing methods.

**Summary Of The Review:**

Overall, I think this is a solid paper and it encourages future work on compressing training sets into queryable encodings. The key contribution seems to be the computational efficiency of their method. It's clear that it's more computational efficient at the same performance, but it's not very clear how important computational efficiency is here.

---

> ### Author Response · Authors · 2022-11-19
> **Response to Reviewer GJez (Part 1)**
>
> We thank the reviewer for their detailed feedback. We address the reviewer's concerns below.
>
> **Concern 1: The motivation for why we need more compute-efficient methods than NPT is not clear.**
>
> *Response 1.1. We need more compute-efficient methods because NPT does not perform well in settings that require supporting a large context.*
> SPIN can be applied to problems to which NPT cannot be easily applied (e.g., genotype imputation). Such problems require models that support a large context size. Table 4 provides an example of a genotype imputation problem on which NPT does not run while SPIN does. Under Response 1.3 below, we include an additional genotype imputation experiment on which NPT performs much worse than SPIN.
>
> *Response 1.2. We need more compute-efficient methods because neural process (NP) models are more accurate with large context sizes, and SPIN/IPNP supports large context sizes.*
> Conditioning on a larger context size provides additional signal to a meta-learning model, and improves its accuracy. However, existing meta-learning models have an $O(n^2)$ computational complexity as a function of the size of the context $n$. Our methods feature $O(n)$ complexity, and thus support larger contexts. When using these larger contexts, we observe improved performance. In our experiments, we show that by leveraging larger contexts, SPIN and IPNP outperform NPT and other methods (Table 2 of UCI experiment, Figure 3 with meta learning experiment, Table 4 with genomics imputation experiment).
>
> *Response 1.3. A new genomic experiment where SPIN outperforms NPT by virtue of being more compute-efficient.*
> We provide the following genomic imputation experiment where the reference haplotypes in the training set are gradually increased from a small fraction of 1% to 100% of all available haplotypes. Pearson $R^2$ is reported cumulatively for 10 randomly selected regions with window size=300. As the size of genomic biobanks increase, there is a growing need for methods that can scale.
>
> The table below shows that the performance for both SPIN and NPT improves with increasing size of the reference dataset. However, NPT cannot be used beyond a certain set of reference samples due to its GPU memory footprint.
>
>
> |  Reference Samples|| 1% (44)|5% (219)|15% (658)|30% (1316)|100% (4388)|
> |:-:|:-:|:-:|:-:|:-:|:-:|:-:|
> |  Pearson $R^2$ $\uparrow$| SPIN|84.87|86.25|87.55|90.33|92.91|
> |  | NPT |85.00|85.54|86.35|-|-|
> | GPU Mem(GB)| SPIN|8.64|8.77|9.10|9.59|12.83|
> |  | NPT|18.07|18.43|19.69|OOM|OOM|

---

> > ### Author Response · Authors · 2022-11-19
> > **Response to Reviewer GJez (Part 2)**
> >
> > **Concern 2: The need for intuition for folding and for the XABD and ABLA modules**
> >
> > *Response 2.1. Intuition behind the folding operation.*
> >
> > We use the term unfolding to refer to a reshaping operation where the tensor of inducing points is reshaped from ($h,f,e$) to ($1,h, f \cdot e$); the folding operation is defined as going from ($1,h,f \cdot e$) to ($h,f,e$). Unfolding can be viewed as transforming each of the $h$ inducing points from a sequence of $f$ embeddings of size $e$ into a single vector of size $f \cdot e$. Folding undoes this transformation. Representing inducing points and regular datapoints as vectors of size $f \cdot e$ enables us to perform cross-attention between them.
> >
> > The unfolding operation is key to the XABD operation, since we are attending between the induced datapoint latent $h$ from unfolded $H_D$ (with shape ($1,h,f \cdot e$)) and the datapoints $n$ from unfolded latent $H_A$ (with shape ($1,n,f\cdot e$)).
> > The folding operation is used to get back to the original shape of $H_D$.
> >
> > *Response 2.2. Intuition for the ABLA module.*
> >
> > We draw inspiration for the ABLA module from existing literature, where self-attention is performed on projected latents. ABLA similarly performs self-attention on the $H_A$ latent and thus has low computational cost. Our reasoning for including the ABLA was to provide a general purpose architecture that could be used to benefit various datasets.
> >
> > *Response 2.3. Intuition behind the XABD module.*
> >
> > Intuitively, SPIN tries to "compress" a large dataset $\mathcal{D}$ into a smaller set of inducing points $\mathcal{H}$. Often, many datapoints in $\mathcal{D}$ will be similar to each other, and thus redundant. Inducing point methods replace such redundant datapoints with a single inducing points. Classical inducing point methods rely on an optimization procedure to find $\mathcal{H}$. In contrast, SPIN uses the XABD layer to compute $\mathcal{H}$ from $\mathcal{D}$ using cross-attention. The XABD layers takes as input a tensor of inducing points $\mathbf{H}$ and a tensor of datapoints $\mathbf{D}$; it outputs a new tensor of inducing points $\mathbf{H}'$ that are formed when $\mathbf{H}$ attends to $\mathbf{D}$; each inducing point in $\mathbf{H}$ queries $\mathbf{D}$, attends to a group of similar datapoints, and updates its representation in a way that summarizes the signal contained in this group of datapoints.
> >
> > In order to better explain the workings of the XABD module, we perform an additional experiment, which demonstrates that the above intuition is correct (see Appendix A.8).

---

> ### Author Response · Authors · 2022-12-08
> **Follow-up**
>
> We again thank the reviewer for the time and expertise invested in the review. We would like to kindly ask the reviewer to let us know whether our answers and changes to the manuscript have clarified the raised concerns, and whether this changes the reviewers evaluation of the paper. Thank you, the authors.

---

### Official Review · Reviewer_jnqW · 2022-11-04

**Confidence:** 3
**Correctness:** 4
**Technical Novelty And Significance:** 3
**Empirical Novelty And Significance:** 3
**Recommendation:** 6

**Clarity, Quality, Novelty And Reproducibility:**

The paper is very well written and easy to follow.  I think the presented architecture is novel, but I’m a little unsure given how many transformer / attention variants there are currently.  So I would defer to other reviewers on novelty.  The authors provide detail about their architectures and in the appendix they detail hyperparameters, so I’d feel confident I could reproduce the experiments with some effort.

Nitpicking:
Inducing points are not attributable to Wilson et al.  That’s a strange citation to have. You should cite e.g. Snelson & Ghahramani, 2005.  I’d really like to see a better treatment of the existing literature on inducing point approximations.

“Inference” is an overloaded term in deep learning and statistical inference.  I.e. they mean different things in deep learning and non-parametric modeling.  I’d just avoid using it altogether.



**Strength And Weaknesses:**

Strengths:
The paper is clear and well written.
The proposed model seems intuitive, scalable and effective.
I like that the authors found multiple problem domains where their proposed approach would outshine traditional GP or deep learning approaches.  Genomic imputation is neat and seems like an important real world application.
The method seems to work well in the experiments presented.
The GPU memory required by the method seems clearly less than the stronger baseline NPT.

Weaknesses:
A major aspect of the neural process literature is that the models are a stochastic process, i.e. they induce a distribution over functions conditioned on some observed data.  This paper completely ignores that part and considers only predictive accuracy.  For example, a major claimed advantage of neural processes is that they can provide a high quality estimate of uncertainty away from the data.  This paper completely ignores the “processes” part of neural processes and evaluates the models as if they’re standard deep networks on standard deep learning benchmarks (agreed genomic imputation is non-standard and more exciting).
The treatment of existing work is a little sloppy.  I’m not sure why inducing points and deep kernel learning are attributed to Wilson & X.  There’s a deep literature on both subjects and these papers are one contribution of many in each subfield (neither the first, last or SOTA).
The choice of baselines seems quite weak.  Particularly, UCI has been completely and utterly annihilated by Bayesian deep learning papers.  Take a look at all the papers citing Bayesian dropout (Gal & Gharhamani).  The fact that the baselines here are: a method from 2001 - GBT, K-Nearest Neighbors and “Multilayer Perceptron” seems quite suspect given that one can just extract the numbers from recent papers.  Why are all the baselines from 20 years ago or more?  That is compounded by the fact that only a ranking is provided, so a reader can’t directly compare to more recent literature.  I found the results in the appendix and confirmed that 1) in general they’re not as strong as dropout (which is taken to be a straw man in many papers) and 2) the results often don’t seem statistically significant.  Could the authors provide some more context about why they chose these baselines and the significance of the ranking results?
The ranking results are just not statistically significantly better than NPT.  I think it’s ok to say they are competitive with NPT but require less than half the memory, but it doesn’t seem justified to say that they are better on average.
If the claim is that SPIN / IPNP is faster, then plot with walltime as an axis.


**Summary Of The Paper:**

This paper develops a special kind of neural network architecture that the authors call SPIN, semiparametric inducing point networks.  This architecture takes a dataset as input and using a
“dataset encoder” outputs a latent set of “inducing points” which one could imagine either as pseudo-examples representing the larger dataset or summary statistics describing the data.  These inducing points are incorporated into a prediction network via cross-attention layers that effectively compute dot products between a query and the inducing points.  The authors demonstrate the efficacy of this approach empirically on the UCI data, a meta learning setup and a genotype imputation problem.

**Summary Of The Review:**

Overall, I think this paper represents an interesting innovation that seems novel and is well described.  I think the paper missed the mark a bit in terms of formulating / evaluating the method as a stochastic process.  The experiments are useful as a proof of concept and the genotype imputation experiment presents a scenario where this model seems to make sense over standard deep learning approaches.  The UCI experiments are a bit weak in terms of the baselines and the overall results.  Overall I think the novelty and presentation warrant an accept but the paper could be stronger.  I would vote accept but I wouldn’t champion the paper strongly (e.g. for a presentation).

---

> ### Author Response · Authors · 2022-11-19
> **Response to Reviewer jnqW (Part 1)**
>
> We thank the reviewer for their detailed feedback. We address the reviewer's concerns below.
>
> **Concern 1: The paper needs to analyze the proposed models from a probabilistic and stochastic process perspective**
>
> *Response 1.1. Our paper introduces IPNPs as probabilistic models in a manner consistent with the NP literature.*
> Specifically, IPNPs are defined as a distribution over target labels conditioned on target dataset inputs and context data; see, e.g., Section 2:
> > “... a distribution on target labels $\mathbf{y}_t \sim p(\mathbf{y} \mid \mathbf{x}_t, \mathbf{r}_c(\mathcal{D}_c)),$ where $p$ is a probabilistic model...”
>
> We also make this formulation explicit in Section 3.3 where we describe IPNPs and in Tables 15 and 16 in Appendix A.6 where we provide the architecture used for IPNPs and CIPNPs. These show that our decoder outputs distributions over labels parameterized by learned parameters $\mu$ and $\Sigma.$ Note also that IPNPs admit latent variables $z$, just like standard neural processes---we call these models CIPNPs.
>
> *Response 1.2. We evaluate IPNPs using standard metrics for probabilistic models.*
> The log likelihood metric that we report in the results of the NP experiment in Section 4.2 also quantifies the uncertainty of our NP models and is widely used in the NP literature (see, for example, Figure 3 in [1]). This metric, which we also use to train our models, incentivizes NP models to output distributions such that the ground truth target labels observed at training time have large density.
>
> *Response 1.3. Additional experiments that highlight the probabilistic and stochastic process nature of IPNPs.*
> To further draw the connection between inducing point NPs and uncertainty estimation, we have performed additional experiments:
> * *Additional Experiment 1.3.1. Appendix A.6: Qualitative Uncertainty Estimation.* We present a new figure that shows how our NP models also output distributions over function space and that CIPNP and IPNP models better capture uncertainty in areas where context points have not been observed.
> * *Additional Experiment 1.3.2. Appendix A.6: Quantitative Calibration Results.* The figure and table in this section demonstrate that in lower context regimes inducing point NPs are better calibrated.
>
> We also note that in all of our experiments, CIPNPs---that is, IPNPs that feature latent variables $z$---achieve the best performance, which further highlights the value of using uncertainty-aware probabilistic models
>
>
> [1] Hyunjik Kim, Andriy Mnih, Jonathan Schwarz, Marta Garnelo, Ali Eslami, Dan Rosenbaum, Oriol Vinyals, and Yee Whye Teh. Attentive neural processes. In International Conference on Learning Representations, 2018.
>
>
> **Concern 2: The need for additional metrics and baselines on the UCI datasets, specifically Bayesian baselines and kernel methods**
>
> *Response 2.1. We have added new baselines on the UCI experiments.*
> The table below provides additional results for Gaussian process, Bayesian ridge regression and kernel ridge regression. Additionally, our paper compares against baselines used by NPT, which is the most relevant prior work in this area. We are currently running additional baselines that we hope to release over the course of the discussion period.
>
>
> *Response 2.2. Our choice of metrics and evaluation methodology mirrors that of NPT (Kossen et al., 2021), which is the most relevant prior work.*
> Since the UCI datasets comprise of both classification (6) tasks and regression (4) tasks, ranking provides a way to evaluate performance across different metrics and tasks. We do provide individual task performance with standard deviation as well as ranking by classification and regression separately in Appendix A.4 and A.5
>
>
> ||  UCI Datasets | SPIN&emsp;&emsp;&emsp;&emsp; |Gaussian&emsp; Processes&emsp; | Bayesian Ridge Regression | Kernel Ridge Regression |
> |:-:|:-:|:-:|:-:|:-:|:-:|
> |Regression  (RMSE$\downarrow$)|Yacht  |1.28 $\pm$ 0.66 |1.71 $\pm$ 0.61|1.38 $\pm$ 0.46|1.32 $\pm$ 0.44|
> ||Boston Housing |3.01 $\pm$ 0.55|3.43 $\pm$ 1.28| 4.64 $\pm$ 0.98|4.62 $\pm$ 1.02|
> || Concrete| 5.17 $\pm$ 0.87|7.33 $\pm$ 0.50| 11.05 $\pm$ 0.66|11.05 $\pm$ 0.67|
> |Classification (Acc.$\uparrow$)|Breast Cancer|96.32 $\pm$ 1.54|96.67 $\pm$ 1.99|NA|NA|
> ||Kick|90.06|OOM|NA|NA|
> ||Income|95.6|OOM|NA|NA|
> ||Higgs|80.01|OOM|NA|NA|
> ||Forest-Cover|96.11|OOM|NA|NA|

---

> > ### Author Response · Authors · 2022-11-19
> > **Response to Reviewer jnqW (Part 2)**
> >
> > **Concern 3: The performance of SPIN does not represent a statistically significant improvement over NPT (Kossen et al.)**
> >
> > *Response 3.1. SPIN uses significantly less memory than NPT, hence can be applied to problems on which NPT does not run.*
> > The main benefit of SPIN relative to NPT is that SPIN uses less memory. As a result, it can be applied to problems to which NPT cannot be applied (e.g., genotype imputation). We also show that by using an increased context size, SPIN and IPNP outperform NPT and other methods (Table 2 of UCI experiment, Figure 3 with meta learning experiment, Table 4 with genomics imputation experiment).
> >
> > *Response 3.2. We added a new genomic experiment where SPIN outperforms NPT.*
> > We added a new genotype imputation experiment to further demonstrate that SPIN yields significant performance improvements as context size increases. We have added this experiment in Appendix section A.3 of the revised paper. The reference haplotypes in the train dataset are gradually increased from a small fraction of 1\% to 100\% available. We observe that the performance for both SPIN and NPT improves with increasing reference dataset size. However, NPT cannot be used beyond a certain set of reference samples due to its GPU memory footprint, while SPIN yields improved performance.
> >
> > |  Reference Samples|| 1% (44)|5% (219)|15% (658)|30% (1316)|100% (4388)|
> > |:-:|:-:|:-:|:-:|:-:|:-:|:-:|
> > |  Pearson $R^2$ $\uparrow$| SPIN|84.87|86.25|87.55|90.33|92.91|
> > |  | NPT |85.00|85.54|86.35|-|-|
> > | GPU Mem(GB)| SPIN|8.64|8.77|9.10|9.59|12.83|
> > |  | NPT|18.07|18.43|19.69|OOM|OOM|
> >
> >
> > *Response 3.3. At small context sizes, the reviewer correctly points out that SPIN attains the same accuracy as NPT, but uses less memory.*
> > Our claim is that the performance of SPIN in terms of predictive accuracy is similar to baseline methods including Set Transformer and NPT when the context size is the same. This comparable performance is achieved using a fraction of the GPU memory.
> >
> > *Response 3.4. We added the following citations for inducing point approximations
> >
> > [2] Michalis Titsias. Variational learning of inducing variables in sparse gaussian processes. In International Conference on Artificial Intelligence and Statistics, 2009.
> >
> > [3] Edward Snelson and Zoubin Ghahramani. Sparse gaussian processes using pseudo-inputs. In Advances in Neural Information Processing Systems, 2005.
> >
> > [4] Trefor W. Evans and Prasanth B. Nair. Scalable gaussian processes with grid-structured eigenfunctions. In International Conference on Machine Learning (ICML), 2018

---

> ### Author Response · Authors · 2022-12-08
> **Follow-Up**
>
> We again thank the reviewer for the time and expertise invested in the review. We would like to kindly ask the reviewer to let us know whether our answers and changes to the manuscript have clarified the raised concerns, and whether this changes the reviewers evaluation of the paper. Thank you, the authors.

---

### Author Response · Authors · 2022-11-19
**Response to Reviewers (General comments)**

Our paper introduces SPIN, a neural inducing point architecture that can query the training set at inference time in a compute-efficient manner. SPIN forms the basis of the inducing point neural process (IPNP), a probabilistic model that achieves state-of-the-art performance on several meta-learning tasks by virtue of being able to support larger training set contexts.

We are glad that the reviewers found our paper clearly written, well-motivated, and having convincing experiments. Based on the helpful feedback from the reviewers, we have revised our paper with the following:
- New results that highlight the probabilistic aspects of IPNPs and demonstrate their ability to model uncertainty better than existing neural process (NP) models (Appendix A.6).
- New genotype imputation experiment with varying context sizes that further motivate the need for meta-learning algorithms that scale to large contexts (Appendix A.3).
- A qualitative analysis that provides additional intuition for the role of the XABD layer within the SPIN architecture (Appendix A.8).
- We replaced all the figures with their vectorized (.svg format) versions.
- We improved the wrapping of tables/figures for better readability.
- Additional citations for multiple papers within the inducing points literature.

We discuss each of these in detail in our responses below. We thank all the reviewers for their service.

---

### Decision · Program_Chairs · 2023-01-20

**Decision:**

Accept: poster

**Justification For Why Not Higher Score:**

Computational gain is significant, however the proposed SPIN approach does not robustly out-perform NPT which is the paper's main baseline.

**Justification For Why Not Lower Score:**

Reviewers all recommend accept in their review. The computational gain of SPIN over NPT is worth presenting at the conference.

**Metareview: Summary, Strengths And Weaknesses:**

This paper proposes semi-parametric inducing point networks (SPIN) as an instance of neural processes. The major innovation here is the design of inducing points and attention blocks to achieve linear run-time cost as the number of datapoint grows.

Reviewers found that the paper is well written, the genotype imputation application is interesting, and the computational gains of SPIN over NPT could be useful in certain practical scenarios.

However, the SPIN approach doesn't seem to provide consistent performance gains over the baseline.

Overall, I think the improved attention architecture with computational gains is useful for the field of neural processes and set data modelling, and I believe researchers in this field can gain some insights from reading this paper.

As a side note: as AC I cannot see the authors' replies to the reviewers (perhaps because the Reader fields were incorrectly specified), so I reached out to ask the authors' reply regarding improvements over the previous submission version of this paper.

**Note From Pc:**

if the above contains the word "oral" or "spotlight" please see: "oral" presentation means -> notable-top-5% and "spotlight" means -> notable-top-25%. As stated in our emails, we are disassociating presentation type from AC recommendations

**Summary Of Ac-Reviewer Meeting:**

N/A